# A diverse proteome is present and enzymatically active in metabolite extracts

Rachel (Rae) J. House[1,2,3,6], Molly T. Soper-Hopper[3,6], Michael P. Vincent[3], Abigail E. Ellis[3], Colt D. Capan[3], Zachary B. Madaj[4], Emily Wolfrum[4], Christine N. Isaguirre[3], Carlos D. Castello[3], Amy B. Johnson[3], Martha L. Escobar Galvis[5], Kelsey S. Williams[2], Hyoungjoo Lee[3] & Ryan D. Sheldon[3] ✉

Metabolite extraction is the critical first-step in metabolomics experiments, where it is generally regarded to inactivate and remove proteins. Here, arising from efforts to improve extraction conditions for polar metabolomics, we discover a proteomic landscape of over 1000 proteins within metabolite extracts. This is a ubiquitous feature across several common extraction and sample types. By combining post-resuspension stable isotope addition and enzyme inhibitors, we demonstrate in-extract metabolite interconversions due to residual transaminase activity. We extend these findings with untargeted metabolomics where we observe extensive protein-mediated metabolite changes, including in-extract formation of glutamate dipeptide and depletion of total glutathione. Finally, we present a simple extraction workflow that integrates 3 kDa filtration for protein removal as a superior method for polar metabolomics. In this work, we uncover a previously unrecognized, protein-mediated source of observer effects in metabolomics experiments with broad-reaching implications across all research fields using metabolomics and molecular metabolism.

Metabolism is a highly dynamic, interconnected, and diverse set of chemical reactions required for energetic homeostasis, redox balance, macromolecular anabolism, and cellular signal transduction. Metabolic phenotypes are not only components but also key drivers of physiological and pathological states. Interrogation and experimental manipulation of metabolism has enhanced our understanding of topics including exercise[1], cancer[2–7], immune function[8–10], and others, while paving the way for metabolism-based therapeutics[11]. Methods that reliably and reproducibly assess biologically relevant metabolic phenotypes are central to the continued success of metabolism research.

Mass spectrometry-based metabolomics enables the detection and quantification of hundreds to thousands of metabolites in a single sample. This process, however, is wrought with experimental and practical challenges that must be carefully considered when designing metabolomics workflows[12]. No single method can capture the entire metabolome due to the vast diversity of compound properties (hydrophobicity, polarity, size, etc.) and sample types (tissue, biofluids, cells, etc.). As such, compound coverage in a metabolomics experiment is constrained by the compatibility of each compound with the analytical approach. This has necessitated the development of countless protocols[13–22]. Such methods are often further refined across labs based on project-specific needs and available instrumentation, leading to innumerable variations of metabolomics methodologies in the literature. This ambiguity complicates reporting and peer-review of metabolomics data[23]. Moreover, the effects of many procedural

[1] Department of Cell Biology, Van Andel Institute, Grand Rapids, MI, USA. [2] Department of Metabolism and Nutritional Programming, Van Andel Institute, Grand Rapids, MI, USA. [3] Mass Spectrometry Core, Van Andel Institute, Grand Rapids, MI, USA. [4] Bioinformatics and Biostatistics Core, Van Andel Institute, Grand Rapids, MI, USA. [5] Office of the Cores, Core Technologies and Services, Van Andel Institute, Grand Rapids, MI, USA. [6] These authors contributed equally: Rachel (Rae) J. House, Molly T. Soper-Hopper. ✉ e-mail: ryan.sheldon@vai.org

details, including extraction solvent composition, additives, timing, pH, etc., on metabolomic data quality are not well defined. Ongoing efforts to optimize metabolomics workflows for maximal compound coverage and the faithful transmittal biological phenotypes are essential.

A critical step in metabolomics workflows is the extraction of compounds of interest from the biological matrix. This commonly involves sample homogenization with organic solvents[24–26] to precipitate proteins and other unwanted biomolecules, leaving metabolites in the soluble fraction for analysis. This insoluble fraction from metabolite extracts can be useful for other analyses such as proteomics and RNAseq[27]. However, evidence supporting the extent of protein precipitation is lacking, though it is generally assumed to be complete. A single report demonstrated that 2–6% of total serum proteins remain in the soluble metabolite fraction in an extraction solvent-dependent manner[28]. This poses an unaddressed risk to accurate phenotype detection with metabolomics. If proteins are present in metabolite extracts from sample types with metabolic enzymes (e.g., cells or tissues) then it is plausible that these extracts are, in fact, unintentional bioreactors of enzymes and their substrates. Whether protein carryover in metabolite extracts causes post-extraction metabolite interconversions has not been evaluated and may lead to false phenotype detection that is not biological in origin.

In this work, we sought to evaluate and optimize metabolite extraction approaches for metabolomics. These efforts unveiled a diverse proteome in metabolite extracts across a wide range of sample and extraction types. Strikingly, when analyzed through gene set enrichment analysis, these metabolite extract proteomes are enriched for metabolic function annotations. Moreover, we demonstrate that residual enzymatic activity causes metabolite interconversions in dried then resuspended extracts. Finally, we provide a metabolite extraction modality using post-extraction protein removal for broad-coverage metabolomics. These findings have broad-reaching implications across all fields that use metabolomics and molecular metabolism, such as cancer, immunology, and diabetes research, and related disciplines, like exposomics.

## Results

### Metabolomic responses to extraction water content

Metabolite extraction constrains the analytical scope of metabolomics experiments, and efforts to optimize extraction conditions may lead to approaches that enhance metabolome coverage. To this end, we sought to improve polar metabolite coverage through post-homogenization water addition using cryopulverized and pooled mouse liver tissue as a model system. We reasoned that this would preserve the critical initial protein precipitation achieved with the commonly used 40% acetonitrile, 40% methanol, and 20% water (AMW20)[29–38] extraction approach, which, when followed with a secondary water addition, would improve the solubility of polar metabolites (Fig. 1a). To identify the optimal amount of water addition for polar metabolites, we titrated the amount of water added to the crude AMW20 extract in 5% increments, achieving a final water content between 25% (AMW25) and 60% (AMW60) (Fig. 1a). Extraction water content significantly affected peak areas of 89 of 195 metabolites detected (Fig. 1b) and led to stratification in the first principal component (44.3%) by PCA (Fig. 1c). Raw peak areas for all metabolites detected in this experiment are available in Supplementary Data 1.

Water addition had a profound effect on nucleotide detection and other phosphate containing compounds. Specifically, we noted a water dose-dependent increase in nucleotide triphosphates, with the greatest stepwise increase occurring between AMW20 and AMW30 (Fig. 1d, Supplementary Fig. 1a). This large increase (5- to 8-fold) in nucleotide triphosphate signal over a relatively small range of extraction water content (i.e., 20% and 30%) highlights the dependency of metabolomics experiments on extraction conditions and the need to

tightly control such variables. Other nucleotides were similarly affected (Supplementary Fig. 1a). Control experiments with addition of a second volume of AMW20 solvent confirmed that metabolomic changes were not due to increased extraction volume, and by extension sample:solvent ratio, alone (Fig. 1e and Supplementary Fig. 1b).

Given the effects of water titration on nucleotides, we hypothesized that compound hydrophobicity would predict metabolome responsiveness to extraction water content. As such, we evaluated the change in metabolite response to water (AMW50/AMW20) as a function of LogP, the partitioning coefficient between octanol (positive LogP, hydrophobic) and water (negative LogP, hydrophilic). In support of this, nucleotides, which have increasingly negative (hydrophilic) LogP values corresponding with increased phosphate number (i.e., nTP > nDP > nMP), were increased in abundance as a function of extraction water content (Fig. 1f). Similarly, acyl-carnitines become increasingly hydrophobic (i.e., more positive LogP) with acyl-chain length and their detected abundance decreased with extraction water content (Fig. 1g). Extending this metabolome-wide, we used cheminformatics to broadly examine the relationship between metabolite hydrophobicity and water content-mediated changes in LCMS-measured abundance. Compound SMILES (Simplified Molecular Input Line Entry System) was used to predict LogP values, which ranged from −4.9 (N-Glycolylneuraminic acid; most hydrophilic) to 11.2 (phosphatidyl-choline, PC[16:0_18:1]; most hydrophobic) among detected compounds (Supplementary Fig. 2a). Using these predicted values and the fold-change for each group relative to AMW20, we hypothesized an inverse correlation between LogP and the Log2FC of increasing water content relative to AMW20. Consistent with this, a significant inverse correlation was observed beginning at AMW30 (Fig. 1h). However, instead of becoming more negative with increasing water content, this correlation weakened and became less negative (Fig. 1l, j; Supplementary Fig. 2b). These data collectively indicate that compound hydrophobicity alone does not fully predict compound recovery from increased extraction water content and implicate the influence of compound extrinsic factors.

A surprising observation from these studies was the extraction water-dependent loss of the internal standard, $D_5$-glutamate (i.e., glutamate with five hydrogens replaced with deuterium atoms), which was added at resuspension (Fig. 2a). This internal standard is a technical control used to monitor LCMS performance over thousands of samples and years of instrument operation and is necessarily very stable. Because $D_5$-glutamate is chemically identical to endogenous unlabeled glutamate, we initially suspected that water-addition led to a global decrease in glutamate detection. However, this was not the case (Fig. 2b), and on further interrogation, we noted a concomitant gain of $D_4$-labeled glutamate (Fig. 2c). We then considered if water addition caused a spontaneous hydrogen-deuterium exchange, but another deuterated internal standard, $D_8$-tryptophan, remained unchanged (Fig. 2d). Further, hydrogen exchange is pH dependent, but extract pH was unaffected by water content (Supplementary Fig. 3). These observations collectively suggest that the $D_5 \rightarrow D_4$-glutamate conversion was not due to spontaneous hydrogen-deuterium exchange. Given our findings that non-compound intrinsic factors contribute to extraction water responses (Fig. 1) and water-dependent conversion of $D_5 \rightarrow D_4$-glutamate (Fig. 2), we next considered whether a protein-mediated mechanism could be responsible.

### Proteins remain in the metabolite fraction during extraction

To determine if the observations in Figs. 1 and 2 were due to enzymatic activity, we conducted proteomic analysis on AMW20 and AMW50 metabolomics extracts (Fig. 3a). First, we assessed total protein content by BCA assay, which revealed 11.3 and 15.7 μg of protein per mg of liver, respectively, compared to 153.8 μg per mg of whole liver lysates (Fig. 3b). However, BCA reagents also react with non-proteinaceous peptides, such as glutathione, which is abundant in the liver[39]. To

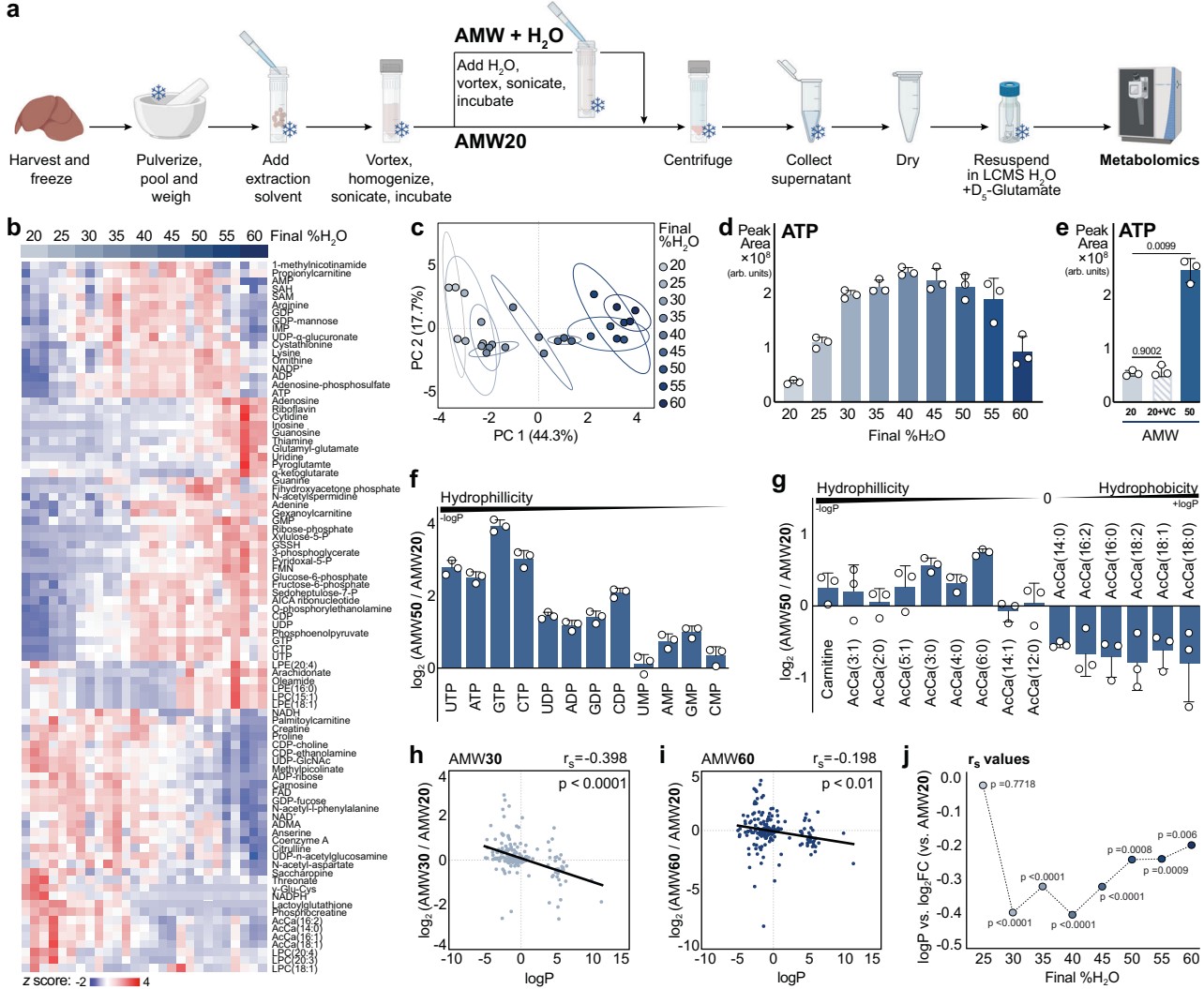

**Fig. 1 | Metabolomic responses to extraction water content are not fully explained by compound hydrophobicity. a** Schematic of AMW and AMW + water addition extractions. **b** Heatmap depicting relative abundance of murine liver metabolites across extraction conditions, from AMW20 to AMW60. All significantly different metabolites are shown. Columns are shaded with a gradient of blue representing the final water percentage. Significance was calculated by One-way ANOVA (alpha=0.05, two-sided, F-stat) with Fisher post-hoc testing using MetaboAnalyst 5.0, and row hierarchical clustering was performed using Morpheus (Broad; metric = 1 minus Pearson correlation; linkage method = average; cluster = rows). **c** PCA of AMW20-AMW60 metabolites in the murine liver (95% confidence ellipse, $n = 3$ technical replicates per group). **d** Peak area abundance ATP content across extraction conditions, from AMW20 to AMW60 (*versus* AMW20, mean ± SD, $n = 3$ technical replicates per group). **e** Relative abundance of detected ATP across AMW20 (20), AMW20 volume control (VC; addition of a second volume of AMW20), and AMW50 (50) extraction conditions. Significance calculated by One-way Welch's ANOVA (*t*-stat, two-sided) with a Dunnett's T3 multiple comparisons test (mean ± SD, $n = 3$ technical replicates per group). **f** Relative abundance of AMW50 nucleotides (*versus* AMW20). Nucleotides are ordered by PubChem XlogP3

(logP) value (mean ± SD, $n = 3$ technical replicates per group). **g** Relative abundance of AMW50 carnitine and acyl-carnitine species (*versus* AMW20). Carnitine and acyl-carnitine species are ordered by ChemAxon LogP (logP) value (mean ± SD, $n = 3$ technical replicates per group). **h** Scatter plot of metabolite logP value *versus* log2 fold change of each metabolite in AMW30 vs AMW20 conditions. Data points represent individual metabolites. Linear regression (black line) was calculated by two-tailed Spearman correlation. Correlation coefficient ($r_s$) and approximate *p*-value, are displayed ($n = 3$ technical replicates per group). **i** Scatter plot of metabolite logP value *versus* log2 fold change of each metabolite in AMW60 vs AMW20 conditions. Data points represent individual metabolites. Linear regression (black line) was calculated by two-tailed Spearman correlation. Correlation coefficient ($r_s$) and approximate *p*-value, are displayed ($n = 3$ technical replicates per group). **j** Summary of regression analyses in Figs. 1h, 1i, and Supplementary Fig. S2b-i demonstrating the relationship between metabolite hydrophobicity and extraction recovery correlation relative to AMW20 in AMW25-AMW60 extraction conditions. Each point represents a two-sided Spearman correlation coefficient ($r_s$) with corresponding approximate *p* value ($n = 3$ technical replicates per group).

distinguish the fraction of BCA signal in AMW extracts arising from bona fide proteins *versus* non-proteinaceous factors, we filtered crude extracts through a 3 kDa filter. Filtration resulted in a 20-30% decrease in calculated BCA protein content compared to unfiltered samples (Fig. 3a, b), with the remaining signal assumed to arise from compounds less than 3 kDa. We subtracted the 3 kDa filter eluate BCA signal from total extract BCA signal and found that 2.7 and 4.1 µg per mg of liver tissue of proteins greater than 3 kDa are present in AMW20 and AMW50 extracts, respectively (Fig. 3c).

Next, we used quantitative proteomics to evaluate the protein composition of AMW20 and AMW50 extracts. Metabolite extracts are widely regarded as being protein-free, yet to our surprise, we detected 1939 and 1760 proteins in AMW20 and AMW50 extracts, respectively, compared to 5177 in whole liver (Fig. 3d). 3 kDa filtration of extracts confirmed the removal of most of these proteins (Supplementary Fig. 4). To better understand the relationship between the extraction water content and the composition of the protein extracts, we added an intermediate AMW extract with 35% final water content (AMW35)

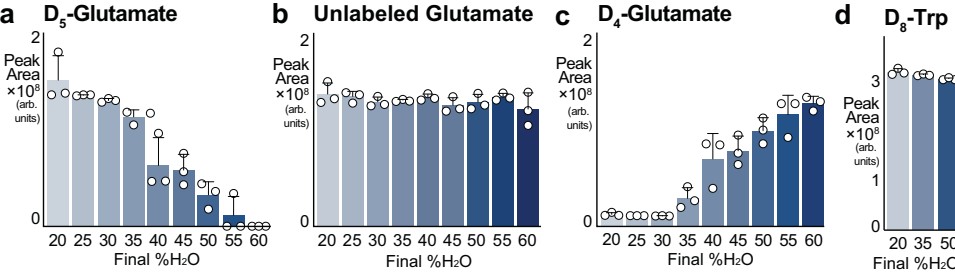

**Fig. 2 | Deuterium loss from D$_5$-glutamate is extraction water dose dependent.** **a** Abundance of D$_5$-glutamate across extraction conditions, from AMW20 to AMW60. D$_5$-glutamate was added at resuspension (mean ± SD, $n = 3$ technical replicates per group). **b** Abundance of unlabeled, sample-originating glutamate across extraction conditions, from AMW20 to AMW60. (mean ± SD, $n = 3$ technical replicates per group). **c** Abundance of D$_4$-glutamate across extraction conditions, from AMW20 to AMW60. D$_5$-glutamate was added at resuspension (mean ± SD, $n = 3$ technical replicates per group). **d** D$_8$-tryptophan is unaffected by extraction water content. D$_8$-tryptophan was added at resuspension (mean ± SD, $n = 3$ technical replicates per group).

and performed proteomics on the extracts. Quantified abundances of all proteins detected in whole liver and metabolite extracts are available in Supplemental Data 2. To understand how extraction water content affects in-extract protein abundance, we performed differential abundance analysis of quantified proteins across the three extraction types. This revealed that proteins were highly responsive to water content, with 1084 significantly affected (Fig. 3e, f). Further, while the majority (1241) of proteins were detected in all three extraction modalities, 161, 187 and 191 proteins were unique to AMW20, AMW35, and AMW50 extracts, respectively (Fig. 3g). To gain further insight into the types of proteins present in metabolite extracts, we performed gene set enrichment analysis (GSEA) against the whole liver proteome. Strikingly, this revealed enrichment of many pathways with metabolic function annotations, including "small molecule metabolic process" as the top pathway (Fig. 3h).

**Enzymatic activity occurs post-extraction**

We then considered whether a protein-mediated mechanism was responsible for the D$_5$ → D$_4$-glutamate conversion (Fig. 2). D$_5$-glutamate contains five deuterated hydrogens, including one on the α-amino carbon, that would be lost through deamination (Fig. 4a). Indeed, eleven transaminases were among the proteins found in metabolite extracts, including the aspartate-glutamate transaminase (GOT1), which was elevated in AMW50 *versus* other conditions (Supplementary Fig. 5a). Transaminases transfer the amine nitrogen of an amino acid to a corresponding ketoacid (e.g., glutamate → α-ketoglutarate (αKG)) using pyridoxal-5-phosphate (pyridoxal-5P) as a cofactor. We also noted water-dependent increases in pyridoxal-5P content (Supplementary Fig. 5b). The concomitant elevation of enzyme and cofactor in AMW50 led us to hypothesize that transaminase activity could be responsible for the D$_5$ → D$_4$-glutamate transition observed in Fig. 2. We monitored the loss of D$_5$-glutamate and appearance of D$_4$-αKG and D$_4$-glutamate using high-resolution LCMS and consistent with glutamate transamination, we observed a loss in D$_5$-glutamate abundance ([M-H] = 151.0774 m/z), and a concomitant gain in D$_4$-glutamate ([M-H] = 150.0711 m/z) in AMW50 *versus* AMW20 (Fig. 4b). The specific loss of the deuterium from the α-amino carbon was supported by MS2 fragmentation of each isotopologue (Supplementary Figs. 6a-c). To causally link this D$_5$ → D$_4$-glutamate shift to sample protein content and specifically to transaminase activity, we repeated this experiment in AMW50 extracts that were either 3 kDa-filtered or in which the pan-transaminase inhibitor aminooxyacetic acid (AOA)[40] was added at extraction. Remarkably, both protein removal and transaminase inhibition with AOA independently preserved D$_5$-glutamate and prevented the appearance of D$_4$-glutamate at 24 hr post-resuspension of dried extracts in water (Fig. 4c).

To gain further insight, we utilized doubly labeled [$^{13}$C$_5$$^{15}$N] glutamate coupled. In this approach, amino acid formation from a

transaminase reaction would only gain the [$^{15}$N] label, and αKG would gain [$^{13}$C$_5$]. In the reverse reaction, [$^{15}$N] labeled amino acids would donate the labeled nitrogen to unlabeled αKG to generate [$^{15}$N] glutamate, or [$^{13}$C$_5$] αKG would be aminated from an unlabeled amino acid to generate [$^{13}$C$_5$] glutamate (Fig. 4d). Note that reverse transamination with [$^{13}$C$_5$] αKG and a [$^{15}$N] amino acid would lead to the regeneration of [U-$^{13}$C$^{15}$N] glutamate, which is indistinguishable from the originally added [$^{13}$C$_5$$^{15}$N] glutamate. Consequently, glutamate labeled only with [$^{15}$N] or [$^{13}$C] underestimates Glu → αKG → Glu futile cycling. Separate [$^{13}$C$_5$] and [$^{15}$N] glutamate species were observed in AMW50 extracts and this was prevented either by filtration or AOA addition (Fig. 4e, f and Supplementary Figs. 6d-f), positively demonstrating a transaminase-mediated futile cycle.

Next, using quantitative LCMS (Supplementary Fig. 7) of glutamate, aspartate, and αKG, we assessed the post-resuspension partitioning of labeled [$^{13}$C$_5$$^{15}$N] glutamate into metabolic neighbors and reincorporation into glutamate. Both 3 kDa filtration and AOA-mediated transaminase inhibition preserved [$^{13}$C$_5$$^{15}$N] glutamate and prevented the formation of [$^{13}$C$_5$] glutamate and [$^{15}$N] glutamate isotopologues (Fig. 4e, f). According to our model (Fig. 4d), transamination of one mole of [$^{13}$C$_5$$^{15}$N] glutamate by transaminases would result in one mole each of $^{13}$C$_5$ and of $^{15}$N products. Indeed, in AMW50 there was ~80-90% decrease from the initially added 82 μM of [$^{13}$C$_5$$^{15}$N] glutamate. This corresponded to an equimolar gain of $^{13}$C$_5$ glutamate with a small contribution of [$^{13}$C$_5$]αKG (1 μM; ~1.3% of [$^{13}$C$_5$] in the system). Interestingly, the formation of [$^{13}$C$_5$]αKG was prevented by 3 kDa filtration, but not AOA. The low αKG to glutamate ratio (~1:100) likely explains this, as minor residual activity or kinetic mismatches (i.e., brief catalytic activity before affected AOA inhibition upon resuspension) would lead to rapid turnover of αKG. Similar to [$^{13}$C$_5$] accounting, we found that the $^{15}$N was distributed between $^{15}$N glutamate (28 μM) and $^{15}$N aspartate (37 μM) (Fig. 4f), both of which were prevented by filtration and AOA. [$^{15}$N] incorporation was not observed in other amino acids (Supplementary Fig. 5c), collectively suggesting prominent glutamate-aspartate transamination. Finally, we summed the concentrations of $^{13}$C or $^{15}$N labeled metabolites in each replicate, which demonstrates that the added [$^{13}$C$_5$$^{15}$N] glutamate is preserved in AMW50F and AMW50 + AOA, and is not lost but rather partitioned between glutamate, αKG, and aspartate in AMW50 (Fig. 4g). Altogether, these results provide clear evidence of enzymatic activity and metabolite interconversions in resuspended metabolomics extracts.

**Proteins in metabolite extracts span sample and extraction types**

The complex interaction between extraction water content and metabolite abundance appears to be driven, in part, by the hundreds of proteins contaminating AMW extracts (Fig. 3) that can be enzymatically active (Fig. 4). We next sought to understand how extraction

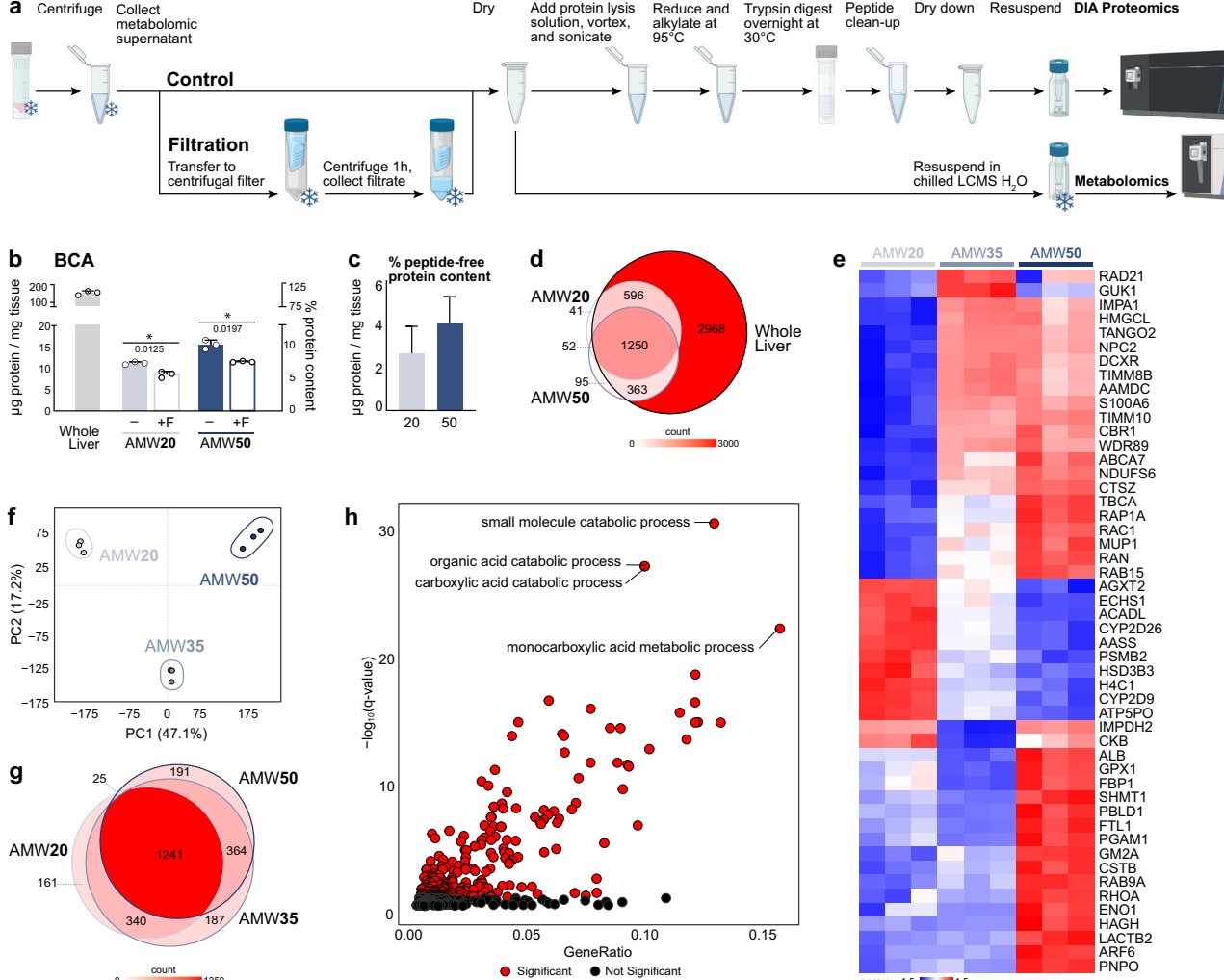

**Fig. 3 | Proteomics of metabolite extracts. a** Schematic of sample preparation post-metabolite extraction for sample filtration, and bottom-up proteomics analysis. **b** BCA determination of protein content normalized to starting tissue amount. Significance calculated by Welch's t-test (two-tailed, $t$-stat, df=2). Percentage protein content calculated as a fraction of the whole liver extract protein content across unfiltered (-) and filtered ( + F) AMW20 and AMW50 samples (mean ± SD, $n$ = 3 technical replicates per group). **c** Peptide free protein content in AMW20 and AMW50 metabolite extractions. Data were log-transformed and analyzed using a linear regression model. Mean differences were calculated using the emmeans package in R. All mean differences are shown on the raw scale. Bar height represents the mean difference between AMW20 and AMW20F or AMW50 vs AMW50F from Fig. 3b. Error bars represents the upper 95% CI for the estimated mean difference. Data are normalized to the average total protein content in a whole liver ($n$ = 3 technical replicates per group). **d** Number of proteins identified in AMW20,

AMW50, and whole liver extracts ($n$ = 3 technical replicates per group). **e** Heatmap depicting relative abundance of murine protein groups quantified across extraction conditions and row standardized. The top 50 most differentially abundant proteins (based on FDR-adjusted p-value) are shown. Significance calculated via LIMMA-eBayes ($t$-stat, two-sided) on Log2 transformed proteins without any missing values ($n$ = 3 technical replicates per group). **f** PCA of AMW20, AMW30, and AMW50 protein groups in the murine liver (95% confidence ellipse, $n$ = 3 technical replicates per group). **g** Number of proteins identified in AMW20, AMW35, and AMW50 metabolite extracts ($n$ = 3 technical replicates per group). **h** Gene set enrichment analysis (biological processes) of proteins significantly impacted by water content. The top 4 pathways (based on FDR q-values) are labelled, and significantly enriched pathways relative to the whole liver metabolome by GeneRatio are colored in red.

modality affects the protein composition of liver metabolite extracts. We selected two additional common metabolite extraction techniques, 80% methanol (MeOH) and the metabolite and lipid-containing aqueous phase (BD-aq) and organic phase (BD-org), respectively, of a Bligh-Dyer extraction (BD; chloroform:methanol:water, 2:2:1.8 v/v), and assessed post-precipitation protein in the soluble phase. Protein content per unit mass of liver, as assessed by BCA, was consistent across metabolite extraction modalities ( ~ 12 μg/mg tissue), and lower in the lipid BD-org fraction (7 μg/mg tissue) (Fig. 5a). Proteomics revealed that a subset of 838 proteins was common to all four extraction types, with 82 being unique to AMW, 6 to MeOH, 206 to BD-aq, and 1050 to BD-org (Fig. 5b). GSEA completed on proteins in metabolite extracts that were differentially abundant between

extraction types revealed, as in (Fig. 2), the strong enrichment of metabolic pathways (Fig. 5c, d). Even the organic layer of a Bligh-Dyer extraction, commonly used for lipidomics, contained >1800 proteins, which were enriched for metabolic function, including purine/pyrimidine metabolism, by GSEA (Supplementary Fig. 8). From GSEA alone, it was unclear whether these enrichments were being driven by metabolic proteins that were more highly abundant in one extraction type but lowly abundant in others. To address this, we examined the relative abundance of proteins across extraction types in the most enriched gene set: "small molecule catabolic process" (Fig. 5e). This revealed a complex interaction between extraction modality and metabolic protein abundance. Thus, metabolic protein carryover in metabolite extracts is not restricted to a specific extraction approach,

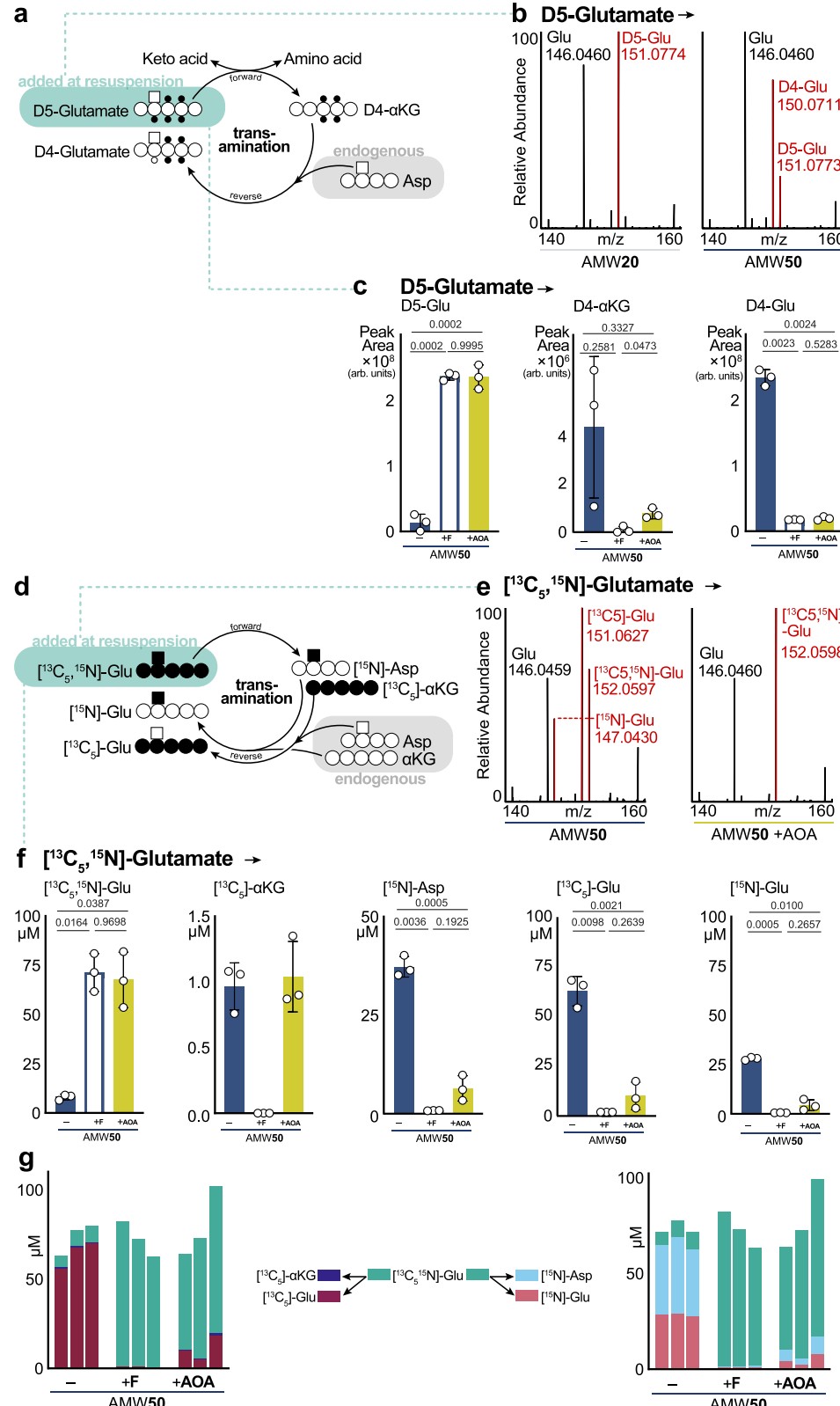

but rather ubiquitous and likely relevant to all metabolite extraction modalities.

Having demonstrated that proteomic content is a ubiquitous feature of metabolite extracts, we interrogated how sample incubation at −80 °C, a common practice in metabolite workflows that is anecdotally believed to aid protein precipitation, would affect proteomic content and enzymatic activity. AMW20 and AMW50 extracts were

incubated in either standard conditions (4 °C for one hr) or overnight (~18 hr) at −80 °C. In the case of AMW50, where water is added to AMW20 extracts after an initial incubation at 4 °C for one hr, we added a fifth condition where water was added after overnight −80 °C incubation (Supplementary Fig. 9a). Remarkably, transaminase activity in AMW50 extracts, as evaluated by D5 → D4 glutamate conversion, was unaffected by extract incubation conditions (Supplementary Fig. 9b).

**Fig. 4 | Active transaminase futile cycling in dried and resuspended metabolite extracts. a** Schematic showing deuterium labeling patterns for a transaminase reaction converting glutamate to α-ketoglutarate and back to glutamate. **b** Mass spectra of glutamate in the murine liver following $D_5$-glutamate addition at resuspension. Representative AMW20 and AMW50 sample spectra are shown. Representative MS1 spectral peaks corresponding to unlabeled glutamate ([M-H] =146.0460 m/z), $D_4$-glutamate ([M-H] = 150.0711 m/z), and $D_5$-glutamate ([M-H] =151.0773 m/z) are shown. **c** Relative abundance of $D_5$-glutamate-derived metabolites in AMW50 (-), AMW50 +filter (+ F), AMW50 + aminooxyacetic acid (+ AOA) murine liver samples. $D_5$-glutamate was added at resuspension in all samples. AOA was added at resuspension in AMW50 + AOA samples. Significance calculated by One-way Welch's ANOVA (t-stat, two-sided) with a Dunnett's T3 multiple comparisons (mean ± SD, $n = 3$ technical replicates per group). **d** Schematic showing $^{13}$C and $^{15}$N labeling patterns for a cyclical transaminase reaction following [U$^{13}$C,$^{15}$N]-glutamate input. **e** MS1 mass spectra of glutamate in the murine liver following [$^{13}$C$_5$,$^{15}$N]-glutamate addition at resuspension. AOA was added at resuspension in AMW50 +

AOA samples. Representative AMW50 and AMW50 + AOA sample spectra are shown. Peaks corresponding to unlabeled glutamate ([M-H] = 146.0460 m/z), [$^{15}$N]-glutamate ([M-H] = 147.0430 m/z), [$^{13}$C$_5$]-glutamate ([M-H] = 151.0627 m/z), and [$^{13}$C$_5$,$^{15}$N]-glutamate ([M-H] = 152.0598 m/z) and are shown. **(f)** Absolute concentrations of [$^{13}$C$_5$,$^{15}$N]-glutamate-derived metabolites in AMW50 (-), AMW50 +filter (+ F), AMW50 + AOA (+ AOA) murine liver samples. [$^{13}$C$_5$,$^{15}$N]-glutamate was added at resuspension. AOA was added at resuspension in +AOA samples. Significance calculated by One-way Welch's ANOVA (t-stat, two-sided) with a Dunnett's T3 multiple comparisons (mean ± SD, $n = 3$ technical replicates per group). **g** Quantitative accounting of the distribution of $^{13}$C$_5$ (left) or $^{15}$N (right) across metabolic neighbors arising from [$^{13}$C$_5$,$^{15}$N]-glutamate added at resuspension. Each bar is an individual replicate arranged in groups of technical triplicates AMW50 (-), AMW50 +filter (+ F), AMW50 + AOA (+ AOA) murine liver samples. The measured concentration of each indicated metabolite isotopologue is stacked to demonstrate conserved molar accounting after metabolic interconversions.

---

Similarly, −80 °C incubation did not affect proteomic diversity or abundance as extract water content was primarily responsible for separation in PCA (Supplementary Fig. 9c). These data indicate that prolonged −80 °C incubation is insufficient to remove proteins from soluble metabolite extracts.

We next asked whether this phenomenon was restricted to the sample type that we selected (liver). The metabolomic and proteomic landscape of metabolite extracts from different sample types (mouse brain, skeletal muscle, perigonadal adipose tissue, plasma, and human HEK cells) extracted with AMW20, AMW35, and AMW50 metabolite extraction modalities were analyzed. Full metabolomics and proteomics data of each sample type are available in Supplemental Data 1 and 2, respectively. In all sample types, metabolite profiling revealed a prominent effect of extraction water content, with water-content separation by PCA occurring in either PC1 (adipose, brain, skeletal muscle) or PC2 (HEK293, plasma) (Supplementary Fig. 10a). In all matrices, we observed an increase in ATP peak area with increased water content, though the magnitude of this increase was matrix dependent (Supplementary Fig. 8b). Skeletal muscle, but not other sample types, showed evidence of in-extract transaminase activity using $D_4$/$D_5$ glutamate as a readout (Supplementary Fig. 10c). We also observed GOT1 and/or GOT2 protein in every sample type except plasma (Supplementary Figs. 10d-e). Using proteomics, a diverse population of proteins were detected in all sample types and, like in the liver, this was strongly influenced by extraction water content (Supplementary Fig. 10f-g). Furthermore, GSEA of each type revealed metabolic gene sets within the top five enrichments for adipose tissue, HEK cells, and skeletal muscle (Fig. 5f). Collectively, these data demonstrate pervasive protein contamination of metabolite extracts spanning multiple extraction and sample types.

**Untargeted metabolomics reveals broad protein-mediated drift**
Given the breadth of proteins contained in metabolite extracts and post-extraction transaminase activity, we pursued a more comprehensive characterization of metabolite interconversions caused by the presence of protein. We hypothesized that post-resuspension enzymatic activity would manifest as time-dependent changes in untargeted metabolomics. Unfiltered and filtered (F) AMW20 and AMW50 liver metabolite extracts were assessed through repeat injections over time after resuspension (approximately every 4 hr over 84 hr) by ESI negative-mode ion-paired LCMS. Of the 330 untargeted features that passed blank and peak-rating filtration, 298 had significant (p < 0.05) time-dependent effects in at least one group, including 89 features common to all groups and 47 features unique to AMW50 (Fig. 6a and Supplementary Fig. 11a). PCA revealed distinct separation of the four groups (AMW20, AMW20F, AMW50, and AMW50F) in PC1 and PC2 (Fig. 6b). Injection-order dependent PCA drift was apparent in each group, occurring primarily in PC2 (22.6%) in AMW20, AMW20F, and

AMW50F, but in PC1 (56.4%) for unfiltered AMW50 extracts (Fig. 6b). This was quantified using the accumulated Euclidean distance in the PCA scores plot from the first to last sample injection in each group, which reveals a ~3x greater magnitude of time-dependent shifts in AMW50 (13.1) versus other groups (AMW20: 5.0, AMW20F: 3.8, AMW50F: 4.8) (Supplementary Fig. 11b). The increased magnitude of time-dependent metabolomics shifts in AMW50 versus other groups is also apparent in plots of the top ten time-dependent features by p-value (Supplementary Fig. 11c).

To evaluate features driving time-dependent PC1 stratification, we conducted PCA of only the AMW50 condition (Fig. 6c). Interrogation of AMW50 PCA loadings (Fig. 6d) showed this time-dependent effect was driven by many features, including an unknown compound (PC1 = +0.073, m/z 279.1137, RT 12.066 min; no formula prediction). Interestingly, this unknown compound coelutes with γ-glutamyl-glutamate dipeptide (Glu-Glu; m/z 275.0886, RT 12.085 min; Supplementary Fig. 12). The mass difference between the unknown and Glu-Glu is 4.0256 Da, which is consistent with the mass shift expected by replacing four hydrogens (1.0078 Da) with deuterium (2.0141 Da) (Fig. 6e). This indicates that the unknown compound is $D_4$-labeled Glu-Glu, arising from the $D_5$ Glu added at resuspension. Both unlabeled and deuterated Glu-Glu were completely absent in other conditions, whereas we noted a time dependent increase in unlabeled Glu-Glu in AMW50, indicating that Glu-Glu is a non-endogenous compound being generated de novo after resuspension (Fig. 6f). We also observed an initial increase in $D_5$Glu-Glu followed by a relative decrease as a fraction of the total Glu-Glu pool (Fig. 6f). This biphasic response of $D_5$ Glu-Glu corresponds with the concomitant, time-dependent conversion of $D_5$Glu to $D_4$Glu (Fig. 6g). Further, with $D_4$Glu and $D_5$Glu accounting for roughly 50% of the total free glutamate pool (Figs. 1, 4, 6g), if Glu-Glu dipeptide production was sourced entirely from the free glutamate pool, then it should yield M + 8 ($D_4$Glu-$D_4$Glu; [M-H] m/z = 283.1376), M + 9 ($D_4$Glu-$D_5$Glu; [M-H] m/z = 284.1439), and M + 10 ($D_5$Glu-$D_5$Glu; [M-H] m/z = 285.1501) deuterium isotopologues that sum to roughly 25% of the total Glu-Glu pool. Instead, none of these higher order deuterium isotopologues (M + 8, M + 9, M + 10) were observed (Fig. 6e). This suggests that only one glutamate in the Glu-Glu dipeptide arises from the free glutamate pool while the other arises from another glutamate source, such as protein hydrolysis, and that it is likely enzymatic since Glu-Glu formation is completely prevented by protein removal.

Another highly influential feature noted from PCA loadings (PC1 = −0.073) had a m/z of 320.0924 and RT 8.792 min (Fig. 6d). This feature had a predicted chemical formula of $C_{11}H_{19}N_3O_6S$ (Δppm = 1.0), and its MS2 fragmentation was highly similar to reduced glutathione (GSH; [M-H] m/z = 306.0765) (Supplementary Fig. 13). The mass difference between the unknown and GSH is 14.0156 Da, consistent with an MS2 neutral loss of a methyl group (CH$_2$). We putatively identified

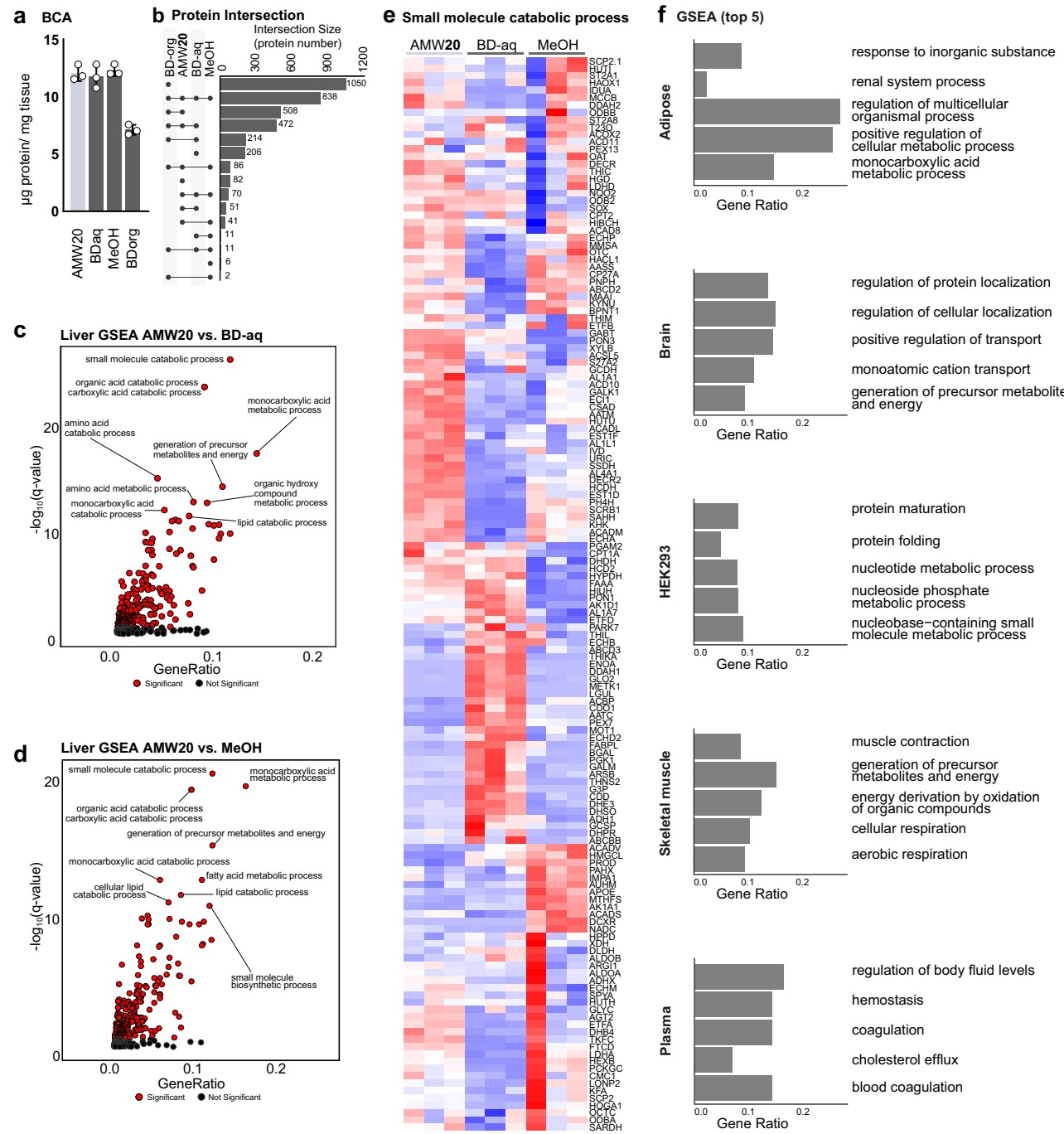

**Fig. 5 | Effects of common extraction modalities on proteomics and metabolomics in multiple sample types. a** BCA determination of protein content, normalized by starting tissue amount across AMW20, Bligh-Dyer aqueous phase (BD-aq), 80% methanol, and BD-organic phase (BD-org) in murine liver extractions (mean ± SD, *n* = 3 technical replicates per group). **b** Upset plot depicting protein intersections between the four extraction conditions (*n* = 3 technical replicates per group). **c** GSEA of BD-aq *versus* AMW20 Biological processes with the top 5 most impacted pathways based on FDR q-values labelled. Gene set enrichment analysis (biological processes) of proteins significantly impacted extraction type. The top 10 pathways (based on FDR q-values) are labelled, and significantly enriched pathways relative to the whole liver metabolome by GeneRatio are colored in red. **d** GSEA MeOH *versus* AMW20 gene set enrichment analysis (biological processes) of

proteins significantly impacted by extraction method. Gene set enrichment analysis (biological processes) of proteins significantly impacted by extraction type. The top 10 pathways (based on FDR q-values) are labelled, and significantly enriched pathways relative to the whole liver metabolome by GeneRatio are colored in red. **e** Heatmap depicting relative abundance of "small molecule catabolic process"-associated proteins across extraction conditions and row standardized. All differentially abundant proteins (based FDR p-value) are shown. Significance calculated via LIMMA-eBayes (*t*-stat, two-sided) on Log₂ transformed proteins without any missing values. **f** Top 5 most extraction water-content impacted biological processes based on FDR q-values within each matrix. Matrices include murine liver, adipose, brain, muscle, and plasma, and human Phoenix-AMPHO (HEK293) cells (*n* = 3 technical replicates per group).

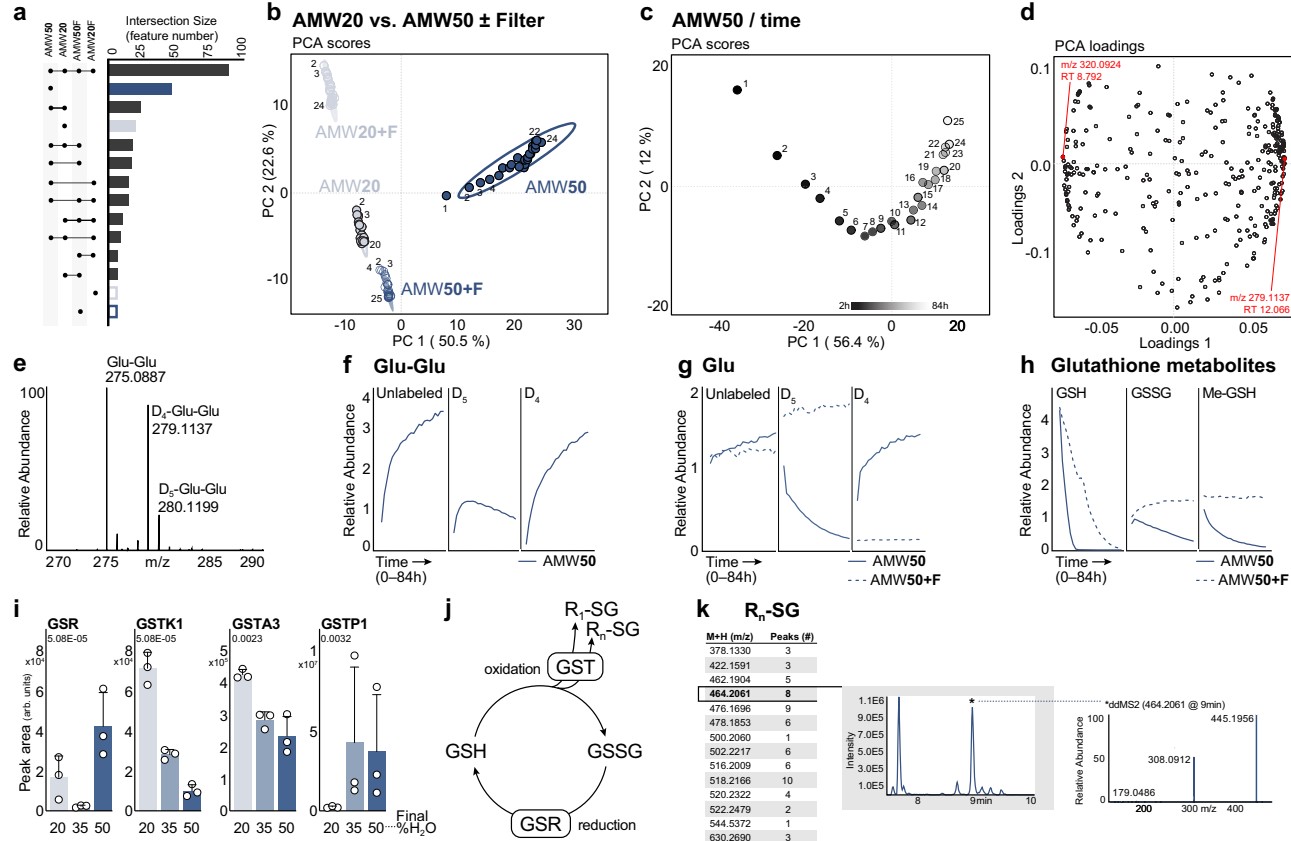

**Fig. 6 | Untargeted metabolomics reveals time-dependent, protein mediated effects in dried and resuspended metabolomics extracts. a** Upset plot showing intersections of features with time dependent drift as defined by significantly non-zero slopes using a generalized additive model. Bars corresponding to individual extraction conditions are colored: AMW20 (light blue, filled), AMW20 + F (light blue, open), AMW50 (dark blue, filled), and AMW50 + F (dark blue, open). **b** PCA of murine liver metabolites from AMW20 ±Filter and AMW50 ±Filter murine liver samples over time (25 repeat injections, 0-84 hr post-resuspension). **c** PCA scores plot of liver metabolites from AMW50 samples showing injection order-dependent metabolite drift (25 repeat injections, 0-84 hr post-resuspension). **d** PCA loadings of murine liver metabolites from AMW50 samples over time (25 repeat injections 0–84 hr post-resuspension). Red dots represent the most positive and negative values along PC1 ( + 0.073, m/z 279.1137 at RT 12.066; no formula prediction and −0.073; m/z 320.0924 at RT 8.792; predicted chemical formula $C_{11}H_{19}N_3O_6S$ Δppm = 1.0). **e** MS1 mass spectra at RT = 12.066 showing γ-glutamyl-glutamate (Glu-Glu) in murine liver following $D_5$-glutamate addition at resuspension. Representative AMW50 sample spectrum is shown. Peaks corresponding to unlabeled Glu-Glu (275.0887 m/z), $D_4$-Glu-Glu (279.1137 m/z), and $D_5$-Glu-Glu (280.1199 m/z) are shown. **f** Relative abundance of $D_5$-glutamate labeled glutamyl-glutamate (Glu-Glu) over time (25 repeat injections 0·84 hr post-resuspension) in AMW50 (solid) samples. **g** Relative abundance of $D_5$-glutamate-labeled glutamate over time (25 repeat injections 0–84 hr post-resuspension) in AMW50 (solid) and AMW50 + F (dashed) samples. **h** Relative abundance of GSH, GSSH, and Me-GSH over time (25 repeat injections 0·84 hr post-resuspension) in AMW50 (solid) and AMW50 + F (dashed) samples. **i** Relative abundance of murine GSR and GSTs in metabolite extract across AMW20, AMW35, and AMW50 extraction conditions. Main effect FDR value shown (mean ± SD, n = 3 technical replicates per group). **j** Schematic showing GSR-mediated glutathione reduction, spontaneous glutathione oxidation, and GST-mediated glutathionylation. **k** Precursor ions with glutathione fragment ion, m/z = 308.0912 using the T3 method. Number of chromatographic peaks identified with a peak width greater than 6 s, and peak intensity 1x10⁵ or greater. Representative chromatogram for precursor ion m/z 464.2061 (± 5ppm), and the ddMS2 for that ion at RT = 9.0 min.

this feature as S-methyl glutathione, which was confirmed with an analytical standard spike-in (Supplementary Fig. 13). The abundance of this compound decreased over time (Fig. 6h), leading us to examine reduced (GSH) and oxidized (GSSG) glutathione pools. In both AMW50 and AMW50F extracts (Fig. 6h), we observed a time-dependent decrease in GSH and a concomitant rise in GSSG, likely due to spontaneous glutathione oxidation. Non-enzymatic degradation of GSH through cysteinyl-glycine bond cleavage is unlikely as previous NMR-based studies demonstrate this reaction exhibits slow kinetics and occurs on the order of weeks[41]. Interestingly, GSH depletion occurred more quickly in AMW50 extracted samples but was not met with a corresponding rise in GSSG (Fig. 6h). Instead, GSSG, like Me-GSH, also decreased over time. The decrease in GSSG in an oxidizing environment suggests an enzymatic mechanism, and indeed our proteomics data revealed the presence of twenty-five glutathione metabolizing enzymes (Supplementary Fig. 14). Included in these is glutathione

reductase (GSR) and sixteen glutathione S transferases (Fig. 6i, Supplementary Fig. 14). Thus, the potential exists for GSSG reduction back to GSH and subsequent conjugation to a variety of compounds and/or proteins (Fig. 6j). This offers a possible explanation as to the source of unlabeled, γ-glutamate in γGlu-Glu (Fig. 6d–f).

To better understand the fate of glutathione in metabolite extracts, we examined untargeted ddMS2 spectra from reversed phase (T3) LCMS in AMW50 and AMW50F samples for precursor ions with the characteristic glutathione fragment ion (m/z = 308.0912). We identified 14 precursor ions that gave rise to 67 distinct isobaric (± 2 ppm) chromatographic peaks (peak width greater than 6 sec, and peak intensity 1E⁵ or greater) in AMW50 samples that were absent in AMW50F samples (Fig. 6k). Though the molecular identities of these glutathionylated compounds remain unknown, these data suggest the presence of a protein-mediated glutathione sink that explains the time-dependent, in-extract loss of total glutathione (Fig. 6h) in AMW50 samples.

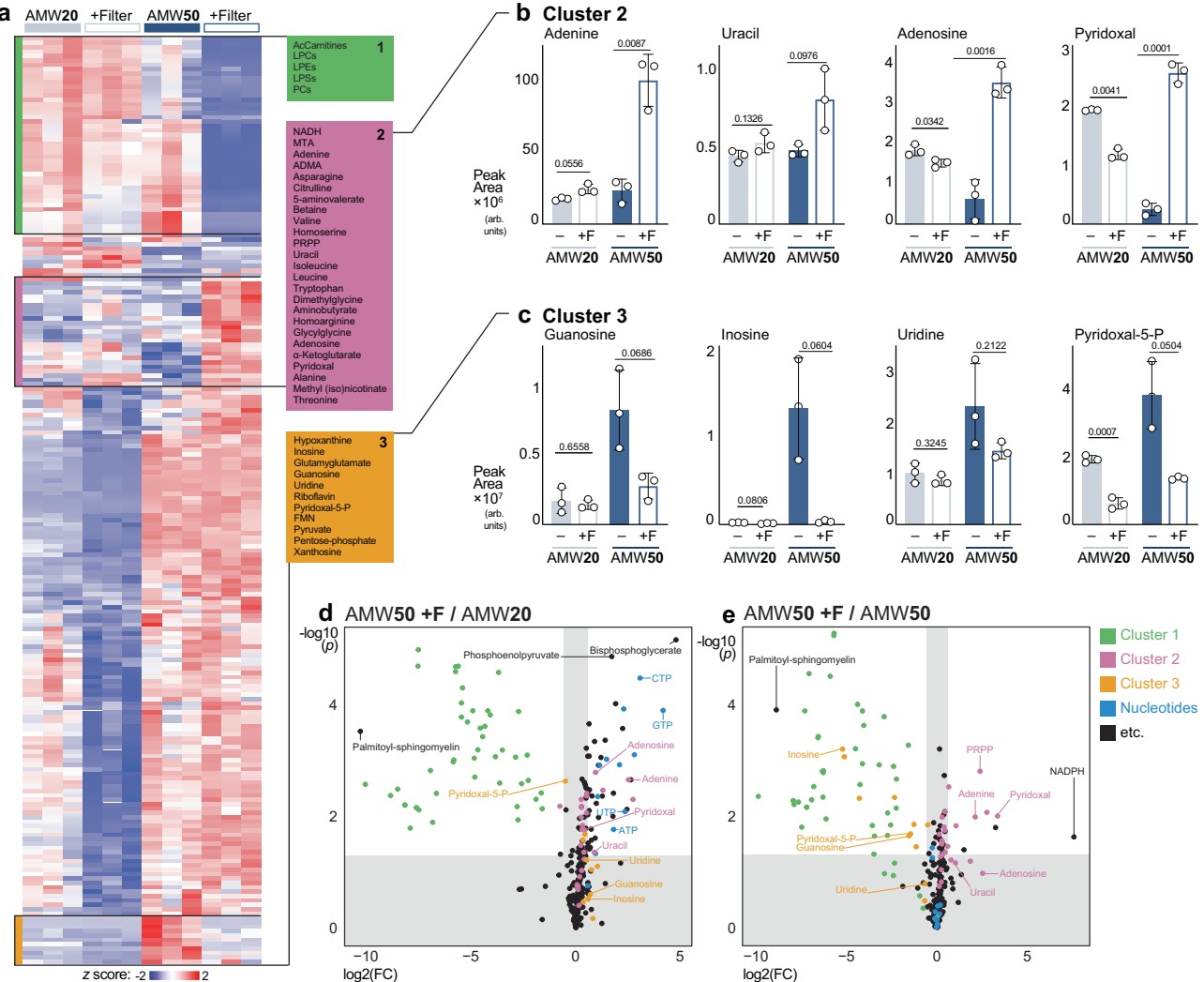

**Fig. 7 | 3kDa filtration in high water content AMW extracts improves metabolite recovery and prevents unwanted protein-mediated effects. a** Heatmap depicting relative abundance of murine liver metabolites across AMW20 ± Filter and AMW50 ± Filter extraction conditions. All significantly different metabolites are shown. Significance was calculated by One-way ANOVA (alpha=0.05, two-sided, F-stat) with Fisher posthoc testing using MetaboAnalyst 5.0, and row hierarchical clustering was performed using Morpheus (Broad; metric = 1 minus Pearson correlation; linkage method = average; cluster = rows). **b** Relative abundance of Cluster 2-related murine liver metabolites across AMW20 ± Filter and AMW50 ± Filter extraction conditions. Significance calculated by One-way Welch's ANOVA (t-stat, two-sided) with a Dunnett's T3 multiple comparisons (mean ± SD, n = 3 technical replicates per group). **c** Relative abundance of Cluster 3-related murine liver metabolites across AMW20 ± Filter and AMW50 ± Filter extraction conditions.

Significance calculated by One-way Welch's ANOVA (t-stat, two-sided) with a Dunnett's T3 multiple comparisons (mean ± SD, n = 3 technical replicates per group). **d** Volcano plot of relative abundance of murine liver metabolites between AMW50 + F and AMW20. All metabolites are shown, metabolites are colored based on cluster and metabolite class, as defined in Fig. 1a. Significance was calculated by t-test with FDR correction (two-tailed, parametric, cutoff = 0.05) using MetaboAnalyst 6.0 (n = 3 technical replicates per group). **e** Volcano plot of relative abundance of murine liver metabolites between AMW50 + F and AMW50. All metabolites are shown, metabolites are colored based on cluster and metabolite class. Significance was calculated by t-test with FDR correction (two-tailed, parametric, cutoff = 0.05) using MetaboAnalyst 6.0 (n = 3 technical replicates per group).

## Water addition with protein removal for polar metabolomics

Given our observations that in-extract water addition improves polar metabolite detection (Fig. 1), but also that it exposes extracts to post-extraction protein mediated metabolic interconversions (Figs. 2, 4, 6), we hypothesized that combining high-water AMW extraction with 3 kDa protein removal would improve polar metabolite coverage while mitigating the risks posed by post-extraction enzymatic activity. We subjected liver samples to AMW20 and AMW50 extractions with or without 3 kDa filtration. Targeted metabolomics was performed on three orthogonal methods as described in Methods. Of the 287 compounds detected, 212 were significantly different by ANOVA and *post hoc* multiple comparisons test. Hierarchical clustering revealed distinct clades of compounds that were affected by filtration (Fig. 7a).

Consistent with Fig. 1, this included elevated levels of polar metabolites, such as nucleotides, in AMW50 that were unaffected by filtration (Fig. 7a and Supplementary Figs. 15a, b). Interestingly, filtration depleted long-chain polar lipid species including acyl-carnitines and lysophospholipids (Fig. 7a, Cluster 1, and Supplementary Fig. 15c) in the AMW50 samples. This is likely due to non-specific binding of lipids to the filter material (regenerated cellulose membranes), rather than because of size exclusion. A second cluster containing many amino acids, was elevated in AMW50F *versus* AMW50 (Fig. 7a, Cluster 2), suggesting that these compounds are present in the original sample but lost through a protein-mediated mechanism specifically in AMW50 extracts. Finally, a third cluster revealed compounds that are elevated in AMW50, but prevented by filtration (Fig. 7a, Cluster 3), including the

de novo formed of glutamyl-glutamate described in Fig. 6e, f. Interestingly, several nucleosides and nucleobases including adenine, uracil, and adenosine are constituents of Cluster 2 (Fig. 7b) while hypoxanthine, inosine, guanosine, uridine, and xanthosine are grouped in Cluster 3 (Fig. 7c). This is consistent with our observations of proteins in purine and nucleoside metabolic pathways enriched in metabolite extracts (Fig. 3, Supplemental Data 2). Similarly, pyridoxal content was diminished (Fig. 7b) while pyridoxal-5P was elevated (Fig. 7c) in AMW50 extracts, which was prevented by filtration. This apparent protein-mediated, in-extract conversion of pyridoxal to pyridoxal-5P may contribute to transaminase activity in these extracts (Fig. 4). Volcano plots of relevant pairwise comparisons (Fig. 7d, e, Supplementary Fig. 15a) are provided to highlight the magnitude of metabolite specific fold-changes to different extraction conditions. For example, AMW50F markedly enhances detection (4- to 25-fold change) of nucleotides and other charged compounds, such as phosphoenolpyruvate and bisphosphoglycerate, over AMW20 (Fig. 7d). When comparing AMW50F *versus* AMW50, in addition to previously mentioned cluster 2 and cluster 3 metabolites, NADPH detection was >200 fold greater in AMW50F (Fig. 7e). This suggests NADPH depletion is at least partly protein mediated in AMW50, rather than from spontaneous oxidation. This, coupled with our observations of glutathione-related metabolites in Fig. 6, raises questions about the veracity of using LCMS metabolomics to assess redox balance. Collectively, these data support the use of high-water AMW metabolite extraction with 3 kDa filtration to improve polar metabolite recovery while preventing enzyme mediated metabolic interconversions. However, we note that this approach would require further optimization to be suitable for analyzing polar lipids.

## Discussion

Here, we present a thorough characterization of the protein composition of metabolomics extracts across common extraction methods and sample types. We demonstrate that metabolite extract proteomes are enriched for metabolic proteins that can retain or recover activity despite exposure to denaturing chemical conditions. Our findings have important implications across all fields utilizing metabolomics, as post-extraction enzymatic activity poses a threat to accurate biological phenotype detection and characterization. We address this problem by presenting an extraction method that expands compound coverage and incorporates passive 3 kDa filtration to remove proteins.

Metabolite extraction is a critical early step in metabolomics workflows as it defines the analytical scope of downstream chromatography-mass spectrometry analysis. The use of 80% polar solvents is widely used for profiling metabolomics studies. This originated with an 80% methanol (aqueous) with EGTA extraction that preserved and maximized recovery of nucleotides[24]. This polar extraction method was refined to 40% methanol, 40% acetonitrile, and 20% water with formic acid addition to more rapidly quench metabolism – namely ATP hydrolysis – and was used for broad metabolite recovery from *E. coli*[25], though formic acid was later shown to be dispensable[20]. This AMW method is widely used for metabolomics studies (see[29–38] for a non-exhaustive list) and was the basis of the current approach. However, despite its widespread use, AMW extraction water content varies widely in the literature[13,19,22,42–46] and has not been systematically evaluated on a broad-metabolite scale. With this as motivation, we titrated extraction water content and observed a strong and complex metabolomic response. As an example, an incremental increase in extraction water content from 20% to 25% induced a 4-fold increase in detected nucleotide triphosphate peak area; this nucleotide effect plateaued at an ~8-fold increase by 40% water. These data underscore the need to tightly control extraction water content, by adjusting extraction solvent volume to sample amount (i.e., tissue mass, cell number, volume, etc.) to account for sample water content. Additionally, we discovered that many metabolites responded to

increased extraction water content in a manner not consistent with their intrinsic hydrophobicity, leading us to consider a protein-mediated effect occurring post-extraction.

The dogmatic view of metabolite extraction with polar solvents is that larger biomolecules such as RNA, DNA, and protein are insoluble. Indeed, the insoluble fraction can be subsequently extracted and analyzed, enabling single-sample multi-'omic workflows[27,47,48]. However, the presence of these molecules in the insoluble fraction does not necessitate their absence from the soluble, metabolite-containing fraction. The present work demonstrates that 2-4% of the total sample protein content remains and comprises over 1000 individual proteins that are enriched for metabolic annotations. Moreover, these proteins are present in common metabolite extraction modalities including AMW20, 80% methanol, and Bligh-Dyer, though specific protein composition varies in each. Remarkably, this was also true for the organic, lipid-containing layer of Bligh-Dyer extracts. This fraction is commonly used for lipidomics, and the impacts of post-extraction lipid-protein interactions should be the focus of future studies. Similarly, broad protein retention was observed across sample types following AMW20 extraction. Collectively, our data demonstrate that metabolites (i.e., potential substrates) and proteins are being co-incubated during extraction and post-resolubilization.

Protein retention in metabolite extracts can lead to post-extraction enzymatic activity and metabolite interconversions (Figs. 1, 3 and 5). Additionally, a diverse protein-metabolite interactome has been described[49] and it is unknown to what extent de novo metabolite-protein interactions will form. Metabolites that become bound to residual proteins will be masked from LCMS detection. It is likely that some of the time-dependent changes observed in protein-replete AMW50 extracts (Fig. 6) are due to protein-metabolite binding rather than catalytic activity. The extent of this in-extract proteome-metabolome interaction warrants further exploration. Moreover, our proteomics data revealed the presence of several proteases and nucleases (Supplementary Data 2), which, if catalytically active, may lead to the liberation of amino acids and nucleosides from protein and RNA/DNA, respectively. Thus, both post-extraction enzymatic activity and metabolite-protein interactions pose a threat to the faithful detection of biological phenotypes. Since the proteome and metabolome are unique to every extraction modality and sample type (Fig. 5), the potential for confounding influence will be different in every scenario and therefore difficult to test, isolate, and observe. Thus, we posit that the prudent course is to take additional steps to ensure the denaturation and removal of proteins from metabolite extracts.

Here, we elected to use 3 kDa filtration to remove proteins from metabolite extracts while preserving the metabolome. Our proteomics and metabolomics data supports the effectiveness of this protocol. A similar approach using a 10 kDa filter was used on dried and resuspended extracts for NMR analysis, citing metabolite instability, with mixed results[50,51]. We have demonstrated enzymatic activity persists in resuspended extracts and thus subjected the soluble fraction of the crude extract to 3 kDa filtration, then dried the eluate for metabolomics. 3 kDa filtration enabled higher aqueous solvent composition and ultimately improved metabolomic coverage while preventing in-extract metabolic conversions. However, this approach adds an extra step to the workflow that takes additional time, can introduce another source for variation, and potential sample loss. Despite these disadvantages, filtration is the least obtrusive and best preserves the metabolome when compared to other protein removal options. We also evaluated overnight incubation of extracts at −80 °C, but this did not affect protein removal (Supplementary Fig. 9). Sample heating is a common approach to denature proteins but will have broad effects on metabolites. For example, glutamine cyclizes to pyroglutamate when moderately heated[52]. Sample acidification can both denature and precipitate proteins, but will also affect the metabolome, and as

demonstrated here with polar-solvent precipitation, completeness is assumed rather than based on empirical evidence.

Metabolite extract supernatants are commonly collected and dried in a vacuum evaporator, under a stream of nitrogen gas, or through lyophilization, then resuspended for analysis. This serves to both concentrate the extracts and to resuspend them in solvent compatible with the analytical approach. For LCMS workflows, resuspension solvent type should match the beginning mobile-phase composition of the chromatography. Thus, for reversed-phase applications where the chromatography gradient ramps from high-aqueous to high-organic composition, samples are commonly resuspended in water. In metabolomics-variations of normal phase methods, including hydrophilic-interaction (HILIC) and amide chromatography methods, samples are resuspended in a mixture of water and solvent (e.g., 50:50 acetonitrile:water) to make the sample compatible with the chromatography while concomitantly resolubilizing high-polarity metabolites of interest. In the present study, our observations of enzymatic activity in resuspended extracts were made exclusively in samples resuspended in water for reversed phased analysis.

Additionally, metabolomics methods rely heavily on retention time for compound identification, which makes chromatographic stability foundational. The unintended introduction of protein to the analytical system may negatively affect the retention mechanisms of the column, leading to retention time shifts, and/or protein buildup at the column inlet may cause pressure increases, peak broadening, and shortening column life. If these proteins elute from the column, then they may interfere with analyte detection through ion suppression or adduct formation. These issues are difficult to evaluate systemically, but removal of unwanted contaminates from analytical chemistry workflows is a guiding principle and one that is relatively easily accomplished in this case through 3 kDa filtration.

Another important implication of this work is the potential for post-extraction changes to isotopologue distributions from stable isotope tracing studies, obscuring biological interpretation. This is apparent in our observations of a $D_5$-$D_4$ glutamate transition that occurred in the absence of changes to the overall glutamate pool (Figs. 2, 6). In this case, in-extract glutamate transamination leads to aspartate and αKG formation (Fig. 4). This, in turn, mixes carbons between Glu and αKG, and nitrogen between Glu and Asp. If occurring in biological tracing experiments, such in-extract transamination could lead to false interpretations. This risk extends beyond the glutamate transamination observed here, as any post-extraction enzymatic activity may lead to tracer mixing between metabolic neighbors. Such activity may not be apparent in metabolite abundance alone and will likely differ across experimental approaches. Thus, 3 kDa extract filtration is an attractive option to ensure protein removal and preservation of in vivo labeling patterns from stable isotope labeling studies.

Our data offer mechanistic insight into the mathematical correction of time-dependent, analyte specific signal drift using periodic injections of pooled samples, which is common practice in metabolomics data processing[53–56]. Given instrument signal stability, changes over time in signal responses of individual metabolites are postulated to be compound intrinsic or otherwise enigmatic. For example, loss of a metabolite over time is thought to be due to compound instability, whereas signal gain over time may be attributed to longer solubilization time requirements in the resuspension solvent. Our data provide another possible mechanism for this phenomenon, that enzymatic activity and/or protein-metabolite binding kinetics may account for time-dependent compound signal drift. In such cases, the removal of proteins by 3 kDa filtration of extracts would stabilize the signal and obviate the need for peak area correction of these metabolites.

Here, we have discovered a diverse proteome in metabolite extracts, the scope of which is determined by a complex interaction between extraction modality and sample type. We further show that in-

extract proteins can be active and drive post-extraction metabolome changes. This constitutes a previously unknown observer effect in metabolomics, putting biological phenotype detection at risk. We provide a practical solution to mitigate this problem by using 3 kDa filtration. Our workflow integrates increased extraction water content with filtration for protein removal and enhanced broad-coverage metabolomics.

## Methods

This study complies with all relevant ethical regulations; mouse procedures were approved by the Van Andel Institute Institutional Animal Care and Use Committee, Grand Rapids, Michigan, USA. All experiments were conducted as described with three technical replicates per condition unless otherwise stated.

### Tissue collection and processing

Wild-type mice (C57BL/6 J) of mixed ages and both sexes were used in this study. Mice were maintained in a humidity and temperature controlled environment in a 12:12 hr light:dark cycle. Mice were anesthetized with an isoflurane vaporizer and whole blood was collected from cardiac puncture into a 1.5 mL tube containing 10 μL of 0.5 M EDTA. Liver, epigonadal adipose tissue, gastrocnemius skeletal muscle, and brain, were then collected. Tissues were immediately snap frozen in liquid nitrogen ($LN_2$) and stored at −80 °C for later processing. Frozen tissues were pooled by type and pulverized with a mortar and pestle in $LN_2$. Cryopulverized pooled tissues were thoroughly mixed, and, careful to avoid sample thawing, weighed into 30-40 mg aliquots into $LN_2$ pre-chilled 2 mL bead mill homogenizer tubes (19-627, Omni). Whole blood with EDTA was kept on wet ice until processing (<60 min) and centrifuged at 4,000 × g for 10 min at 4 °C. Plasma from each mouse was pooled, mixed by vortex, split into 40 μL aliquots, and stored at −80 °C.

### Cell culture

Human kidney epithelial HEK293 Phoenix-AMPHO cells (ATCC CRL-3213), were maintained in Dulbecco's modified eagle medium (DMEM) no phenol red (Gibco), 10% fetal bovine serum (FBS), 5 mL penicillin-streptomycin (Gibco). Cell lines were not tested for mycoplasma contamination. Cells were maintained at 37 °C in a 5% $CO_2$ incubator. For metabolite extraction, cells were seeded at a density of $1 \times 10^6$ per well onto 6-well plate wells. Cells were grown in DMEM, no phenol red (Gibco), 10% FBS, and 5 mL penicillin-streptomycin (Gibco) for 48 hr at 37 °C in a 5% $CO_2$ incubator. Cells were then washed twice with ice-cold NaCl, and plates were frozen on dry ice, and transferred into a −80 °C freezer to be stored until extraction.

### Metabolite extractions

The efficiency of the tissue homogenization is affected by the fractional volume of the sample in the bead mill homogenization tube. Thus, all samples were homogenized using a ratio of 40 mg tissue per mL of solvent to ensure the same degree of homogenization among samples. When testing higher water concentrations, additional water was added after initial AMW20 (40% acetonitrile:40% methanol:20% water) homogenization to achieve the desired final water content. For consistency, this same approach was followed for sample types (cells, plasma) not requiring bead mill homogenization. Homogenization for Bligh-Dyer (2:2:1.8 chloroform:methanol:water) and 80% MeOH extractions were similarly performed at 40 mg per mL. In all extraction types, the same amount of tissue/cell/plasma equivalents was dried and processed for bicinchoninic acid assay (BCA) protein content, proteomics, or metabolomics as indicated.

Metabolites were extracted using one of the following approaches as indicated in text and figure legends. For AMW extraction, ice-cold acetonitrile (ACN, A955-4, Fisher), methanol (MeOH, A456-4, Fisher), and $H_2O$ (W6-4, Fisher) were added to each matrix at a 4:4:2 (v/v/v)

ratio. For 80% MeOH extraction the approach was the same as AMW. For Bligh-Dyer extracts, samples were homogenized in ice-cold 1:1 chloroform (1024441000, Millipore Sigma):MeOH, followed by the addition of 0.9 volumes of water to achieve the final 2:2:1.8 ratio as reported previously[10].

For solid, tissue matrices, extraction solvent volume was determined by (sample mg)/40 mg x (mL solvent). For the Phoenix-AMPHO cell extraction, solvent volume was determined by (1000 μL per smallest sample cell number) x (sample cell number). For plasma extraction, solvent volume was determined by (800 μL per 40 μL) x (sample μL). After solvent addition, extracts were either homogenized for 30 s (tissue) or vortexed for 10 s (biofluids, cells), sonicated for 5 min, and incubated on wet ice for 1 hr. For AMW extracts with greater than 20% water, after the initial 1 hr incubation additional water was added a final water percentage between 25-60% as indicated. These AMW + $H_2O$ samples were vortexed, sonicated for 5 min, incubated on wet ice for an additional 10 min. Following incubation, AMW20 and AMW + $H_2O$ samples were centrifuged at 17,000 × $g$ and 4 °C for 10 min. The supernatants were collected and centrifuged a second time at 17,000× $g$ and 4 °C for 10 min to ensure complete precipitate clearance from the supernatant. 16 mg-tissue-equivalents of aqueous phase supernatant was collected and dried in a vacuum evaporator. In some cases, 8 mg equivalents were dried instead, but resuspension volume was also halved resulting in equivalent tissue concentration in resuspended extracts and on column. The amount of sample equivalents on column was tightly controlled throughout the study.

### pH analysis metabolite extracts
Dried murine liver metabolite extracts from AMW20-AMW60 extraction conditions were resuspended in $H_2O$, vortexed, sonicated for 5 min, and incubated on wet ice for 10 min. Resuspended sample extracts (50 μL) were transferred to a Seahorse XF-96 assay plate (101085-004, Agilent) along with pH standards: pH 2 (SB101 + NaOH, Fisher Chemical), pH 4 (SB101, Fisher Chemical), pH 7 (SB107, Fisher Chemical), and pH 11 (SB115, Fisher Chemical). Samples and standards pH values were measured using a Seahorse XF-96, which is an accurate micro-pH meter down to 25 μL (Supplementary Fig. 5). The amount of sample equivalents per resuspension extract was tightly controlled across samples.

### Protein removal from metabolite extracts by 3 kDa filtration
Amicon Ultra-2 3 K centrifugal filter devices (UFC200324, Millipore) were pre-rinsed with 1.0 mL of LCMS grade water, and centrifugated for 20 min at 4000 × $g$. All eluate and remaining unfiltered water were removed and immediately followed by the addition of metabolite supernatant to prevent membrane drying. Following the initial centrifugation of metabolite extraction (above), supernatant was transferred to the centrifugal filter device, and centrifuged at 4 °C for 60 min at 4,000 × $g$; 8 mg-tissue-equivalents (determined by % volume of initial extraction volume) of eluate was collected and dried in a vacuum evaporator.

### Protein extraction from dried metabolomics samples
Dried metabolomic supernatant extracts were processed using the EasyPep Mini MS Sample Prep Kit (A40006, Thermo Fisher Scientific). Briefly, dried metabolite extract supernatants were resuspended in 100 μL of lysis solution per 16 mg tissue equivalents to extract proteins. Proteins were quantified using the Pierce BCA Protein Assay Kit (23227, Thermo Fisher Scientific). Proteins were reduced and alkylated at 95 °C for 10 min, and samples were digested overnight with Trypsin/Lys-C at 30 °C at a ratio of 10:1 (protein:enzyme (w/w)). Samples were cleaned with kit supplied peptide clean up columns and dried down in a Genevac SpeedVac prior to resuspension for instrument analysis. Samples were resuspended in 50 μL 0.1% formic acid (FA) (LS118-1, Fisher

Scientific) and diluted with 50 μL of 0.1% trifluoroacetic acid (TFA) (LS119-500, Fisher Scientific).

### Preparation of liver samples for proteomics
Tissue samples were homogenized on the Bead Ruptor Elite (19-042E, Omni International) for 30 s in protein lysis solution. Samples were sonicated and clarified via centrifugation and transferred to a Protein LoBind Eppendorf tube. Proteins were quantified using the Pierce BCA Protein Assay Kit (23227, Thermo Fisher Scientific).100 μg of protein was aliquoted to a 1.5 mL screw top tube (72.703.600, Sarstedt) and digested using the EasyPep Mini MS Sample Prep Kit (A40006, Thermo Fisher Scientific). Briefly, sample volume was adjusted to 100 μL with lysis solution, proteins were reduced and alkylated at 95 °C for 10 min, and samples were digested overnight with Trypsin/Lys-C at 30 °C at a ratio of 10:1 (protein:enzyme (w/w)). Samples were cleaned up with kit-supplied peptide clean up columns and dried down in Genevac SpeedVac prior to resuspension for LC-MS/MS. Samples were resuspended in 50 μL 0.1% FA (LS118-1, Fisher Scientific) and diluted with 50 μL of 0.1% TFA (LS119-500, Fisher Scientific).

### Protein quantitation
Proteins were quantified using the Pierce BCA Protein Assay Kit (23227, Thermo Fisher Scientific) following the vendor supplied protocol. Samples were diluted in LCMS $H_2O$ (W6-4, Fisher Scientific). Plates were read at an absorbance of 562 nm using the Synergy LX Multi-Mode Reader and Gen5 software was used for data analysis (BioTek/Agilent). Polynomial regression was used in the Gen5 software to calculate protein concentrations to a protein standard curve after an average blank absorbance subtraction.

### Data-independent acquisition (DIA) LC-MS/MS proteomics
DIA analyses were performed on Orbitrap Eclipse coupled to Vanquish Neo system (Thermo Fisher Scientific). The FAIMS Pro source (Thermo Fisher Scientific) was located between the nanoESI source and the mass spectrometer. 2 μg of digested peptides were separated on a nano capillary column (20 cm × 75 μm I.D., 365 μm O.D., 1.7 μm C18, CoAnn Technologies, Washington, # HEB07502001718IWF) at 300 nL per min. Mobile phase A consisted of LC/MS grade $H_2O$ (LS118-500, Fisher Scientific), mobile phase B consisted of 20% LC/MS grade and $H_2O$ and 80% LC/MS grade acetonitrile (LS122500, Fisher Scientific), and both mobile phases contained 0.1% FA. The LC gradient was: 1% B to 26% B in 51 min, 80% B in 5 min, and 98% B for 4 min, with a total gradient length of 60 min. For FAIMS, the selected compensation voltage (CV) was applied (−45 V and −65 V) throughout the LC-MS/MS runs. Full MS spectra were collected at 120,000 resolution (full width half-maximum; FWHM), and MS2 spectra at 30,000 resolution (FMWH). Both the standard automatic gain control (AGC) target and the automatic maximum injection time were selected. A precursor range of 380-980 m/z was set for MS2 scans, and an isolation window of 50 m/z was chosen with a 1 m/z overlap for each scan cycle. 32% HCD collision energy was used for MS2 fragmentation.

**DIA database search.** DIA data was processed in Spectronaut (version 18, Biognosys, Switzerland) using direct DIA. Data was searched against *Mus musculus* or *Homo sapiens* reference proteomes as appropriate. The manufacturer's default parameters were used. Briefly, trypsin/P was set as digestion enzyme and two missed cleavages were allowed. Cysteine carbamidomethylation was set as fixed modification, and methionine oxidation and protein N-terminus acetylation as variable modifications. Identification was performed using a 1% q-value cutoff on precursor and protein levels. Both peptide precursors and protein false discovery rate (FDR) were controlled at 1%. Ion chromatograms of fragment ions were used for quantification. For each targeted ion, the area under the curve between the XIC peak boundaries was calculated.

## LC/MS Metabolomics

**Metabolomics approach.** Dried metabolomics extracts were first resuspended in 100% water containing 25 μg per mL $D_8$-Tryptophan (DLM-6903-0.25, Cambridge) for analysis using an ion-paired reversed phase chromatography on an Orbitrap Exploris 240 in ESI negative mode (described below). When appropriate for experimental design, final concentrations of 25 μg per mL $D_5$-Glutamate (DLM-556, Cambridge), 25 μg per mL U-$C^{13}N^{15}$ Glutamate (CNLM-554-H-0.25, Cambridge), and 0.5 mM aminooxyacetic acid (28298, Cayman) were also added to the resuspension solvent. Resuspension volumes were varied by sample type to achieve uniform sample-equivalents per volume. This was 80 μg per μL for tissue, 160 μL per μL for plasma, and 6.70E + 3 cells per μL. 2 μL of resuspended sample were injected on column for each method below. For deeper metabolomic coverage, samples were again dried and resuspended in 50% ACN (v/v) and analyzed using two orthogonal chromatographies (Waters BEH Amide and T3, described below) on an Orbitrap ID-X in ESI positive mode. For all experiments, process blanks were analyzed before and after experimental samples. Pooled samples were injected twice before experimental samples for column conditioning, and every 6-10 sample injections thereafter. Data dependent MS2 (ddMS2) data was collected from pooled samples for compound ID. In some experiments, ddMS2 was also collected on experimental replicates to identify group-specific compounds.

**Tributylamine ion paired reversed phase LC/MS.** As previously reported[9,10,27,57], mobile phase A was LC/MS $H_2O$ (W6-4, Fisher) with 3% LC/MS grade MeOH (A456-4, Fisher), mobile phase B was LC/MS grade methanol (A456-4, Fisher), and both mobile phases contained 10 mM tributylamine (90780-100 ML, Sigma), 15 mM acetic acid, and 0.01% medronic acid (v/v, 5191-4506, Agilent). For the re-equilibration gradient, mobile phase A was kept the same, and mobile phase B was 99% LC/MS grade acetonitrile (A955-4, Fisher). Column temperature was kept at 35 °C, flow rate 0.25 mL per min, and the solvent gradient was as follows: 0–2.5 min held at 0% B, 2.5-7.5 min from 0% B to 20% B, 7.5–13 min from 20% B to 45% B, 13–20 min from 45% B to 99% B, and 20–24 min held at 99% B. The analytical solvent gradient was followed by a 16 min re-equilibration gradient to prep the column before the next sample injection that went as follows: 0-0.05 min held at 99% B at 0.25 mL per min, 0.05-1 min from 99% B to 50% B and 0.25 mL per min to 0.1 mL per min, 1-11 min held at 50% B and 0.1 mL per min, 11–11.05 min from 50% B to 0% B at 0.1 mL per min, 11.05–14 min held at 0% B at 0.1 mL per min, 14–14.05 min held at 0% B and increased flow rate from 0.1 mL per min to 0.25 mL per min, and 14.05–16 min held at 0% B and 0.25 mL per min. Data were collected on an Orbitrap Exploris 240 using a heated electrospray ionization (H-ESI) source in ESI negative mode. The mass spectrometer acquisition settings were as follows: source voltage -2500 V, sheath gas 60, aux gas 19, sweep gas 1, ion transfer tube temperature 320 °C, and vaporizer temperature 250 °C. Full scan data were collected with a scan range of 70–800 m/z at a mass resolution of 240,000. Fragmentation data was collected using a data-dependent MS2 (ddMS2) acquisition method with MS1 mass resolution at 120,000, MS2 mass resolution at 15,000, and HCD collision energy fixed at 30%.

**Amide and T3 LC/MS.** Two methods, referred to as Chromatography 1 and Chromatography 2, were alternated using a Thermo Vanquish dual LC coupled to an Orbitrap ID-X. Chromatography 1 used an Acquity BEH Amide analytical column (1.7 μm, 2.1 mm × 150 mm, #176001909, Waters, Eschborn, Germany) combined with a VanGuard pre-column (1.7 μm, 2.1 mm × 5 mm; 186004799, Waters). Mobile phase A consisted of 10% LC/MS grade acetonitrile (A955, Fisher), mobile phase B consisted of 90% LC/MS grade acetonitrile, and both mobile phases contained 10 mM ammonium acetate (73594, Sigma) and 0.2% acetic acid (A11350, Fisher). Column temperature was kept at 40 °C, flow rate 0.4 mL per min, and the solvent gradient was as follows: 0–9 min from 95% B to 70% B, 9-13 min from 70% B to 30% B, and 13-14 min held at 30% B followed by a 20 min re-equilibration gradient to prep the column before the next sample injection. The amide column re-equilibration gradient was as follows: 0-1 min held at 30% B at 0.4 mL per min, 1-3 min from 30% B to 65% B and 0.4 mL per min to 0.8 mL per min, 3-15 min held at 65% B and 0.8 mL per min, 15–15.5 min from 65% B to 100% B at 0.8 mL per min, 15.5–17 min held at 100% B at 0.8 mL per min, 17–17.5 min from 100% B to 95% B and 0.8 mL per min to 1.2 mL per min, 17.5-19.4 min held at 95% B and 1.2 mL per min, 19.4-19.8 min held at 95% B and decreased from 1.2 mL per min to 0.4 mL per min, and 19.8-20 min held at 95% B and 0.4 mL per min.

Chromatography 2 was a reverse phased chromatography using a CORTECS T3 column (1.6 μm, 2.1 mm × 150 mm, 186008500, Waters, Eschborn, Germany) combined with a VanGuard pre-column (1.6 μm, 2.1 mm × 5 mm, 186008508, Waters). Mobile phase A consisted of LC/MS grade water (W6-4, Fisher), mobile phase B consisted of 99% LC/MS grade acetonitrile, and both mobile phases contained 0.1% FA (A11710X1-AMP, Fisher). Column temperature was kept at 40 °C, flow rate 0.3 mL per min, and the solvent gradient was as follows: 0-10 min from 0% B to 30% B, 10-16 min from 30% B to 100% B, and 16-20 min held at 100% B followed by a 14 min re-equilibration gradient. The T3 column re-equilibration gradient was as follows: 0-6 min held at 100% B and 0.3 mL per min, 6-8 min from 100% B to 0% B at 0.3 mL per min, and 8-14 min held at 0% B and 0.3 mL per min.

For both methods, data were collected with an Orbitrap IDX using an H-ESI source in positive mode. For Chromatography 1, the mass spectrometer parameters were: source voltage 3500 V, sheath gas 60, aux gas 19, sweep gas 1, ion transfer tube temperature 300 °C, and vaporizer temperature 250 °C. Full scan data was collected using the orbitrap with scan range of 105-1000 m/z at a resolution of 120,000. Fragmentation was induced in the orbitrap with assisted HCD collision energies at 20, 40, 60, 80, 100% and with CID collision energy fixed at 35%. For chromatography 2, the mass spectrometer parameters were: source voltage 3500 V, sheath gas 70, aux gas 25, sweep gas 1, ion transfer tube temperature 300 °C, and vaporizer temperature 250 °C. Full scan data was collected using the orbitrap with scan range of 105-1200 m/z at a resolution of 240,000. For both methods, ddMS2 data was acquired with MS1 orbitrap mass resolution at 120,000 and orbitrap MS2 mass resolution at 30,000 for a total cycle time of 0.6 s.

**Targeted metabolomics data analysis.** Peak picking and integration was conducted in Skyline (version 23.1.0.268) using in-house curated compound data bases of accurate mass MS1 and retention time derived from analytical standards and/or MS2 spectral matches on each chromatography method (Supplemental Data 3). Full scan raw data files for all samples of a given experiment were imported and metabolite peaks were auto-integrated based off method-specific, in-house curated compound databases that included molecular formula, precursor adducts, and explicit retention times collected from on-method analyzed chemical standards. Manual peak integrations were performed as necessary to account for any tailing peaks, over-lapping peaks, and minor retention time shifting (if any). If manual integration was performed, the "synchronize integration" function was utilized such that all metabolite integration windows were identical between all sample files. Peak areas were then exported for further data processing and analysis. Data were Log-transformed prior to performing statistical analysis. In cases where the same compound is detected on multiple methods, the method with the lowest relative standard deviation in the pooled sample was selected for that compound. Significant differences were determined by ANOVA with a 5% FDR.

**Structural characterization of glutamate isotopologues.** To gain structural insights into glutamate isotopologues, ddMS2 acquisition targeting each isotopologue was acquired using ion-paired LCMS on an Orbitrap Exploris 240 as described above. Mass spectrometer

acquisition settings were as follows: source voltage -2,500 V, sheath gas 60, aux gas 19, sweep gas 1, ion transfer tube temperature 320 °C, and vaporizer temperature 250 °C. Fragmentation data was collected using a data-dependent MS2 (ddMS2) acquisition method with MS1 mass resolution at 120,000, MS2 mass resolution at 15,000, quadrupole isolation window was 0.4 m/z, and HCD collision energy fixed at 15, 30, and 45% for a total of 5 scans per target. Targeted precursor m/z [M-H] were: $^{13}C_5^{15}N$ Glutamate (152.0597), $^{13}C_5$ Glutamate (151.0627), $^{15}N$-Glutamate (147.0429), $D_5$-Glutamate (151.0773), $D_4$-Glutamate (150.071), and unlabeled Glutamate (146.0459).

**Quantitative LCMS metabolomics.** Calibration curves using neat analytical standards for glutamate, α-ketoglutarate, and aspartate were extracted in the same manner as tissue samples (Supplementary Fig. 7). At resuspension, 12.5 µg per mL of $^{13}C_5$ glutamate, $^{13}C_5$ α-ketoglutarate, and $^{13}C_4$ aspartate was added to each calibration curve standard and tissue sample. To do this, the peak area ratio of $^{13}C$ ISTD to unlabeled compound was used to prepare a calibration curve. Regression analysis was completed in GraphPad Prism (v10), and used to interpolate the endogenous concentrations in pooled AMW50F liver samples. Concentrations were then normalized by starting tissue amount. For quantitation of labeled components in AMW50 conditions, where ISTD is readily interconverted, a ratio of labeled compound to unlabeled compound in the sample was calculated and multiplied by the average endogenous concentration in AMW50F. All sample aliquots were from the same pooled liver supply. The validity of this approach is apparent as it was able to accurately measure the known concentration of the added internal standard (Supplementary Fig. 7c).

**Untargeted metabolomics data analysis.** Compound Discoverer (Thermo Fisher, version 3.3) was utilized for untargeted data analysis. Detected features were assigned putative identifications using MS2 spectral library matching to mzCloud and NIST2020 databases. When no MS2 matches were made, MS1 based chemical formula prediction and accurate mass matches to Chemspider were used for lower-confidence tentative identifications. Features without identification were identified as m/z at retention time (RT). The feature/compound list was refined using blank and peak rating filters. Peak areas for putative compounds passing these filters were subjected to statistical analysis as described above.

**Unknown feature identification.** Following AMW20 and AMW50 extractions of liver samples, dried extracts were resuspended in 100% LC/MS grade water and samples were spiked with either methylglutathione (M4139, Sigma) or glutamyl-L-glutamate (228812500, Fisher Scientific) at a final resuspended concentration of 10 µg per mL. A set of un-spiked AMW20 and AMW50 liver samples were included as controls.

**Statistical Analysis**
Differential abundance of proteins was assessed using LIMMA eBayes via the R v4.3 (https://cran.r-project.org/) package *limma*. Only proteins without missing values in a given pairwise contrast were included in this analysis and were Log2 transformed. P-values were multiple testing adjusted via Benjamini-Hochberg FDR corrections[58] across all contrasts in a given assay (e.g., corrected across all 3 sets of concentration comparisons AMW20 *versus* AMW35, AMW20 *versus* AMW 50, and AMW35 *versus* AMW50). To assess proteins with missing data, missing values were imputed with the arbitrary low value of one. This imputation was chosen as missing data was assumed to be missing due to being below the limit of detection and thus these proteins would rank lower than observed abundances. These imputed data were then analyzed via semi-parametric ordinal regression from the R package *ordinal*[59]. P-values for these models were calculated via likelihood ratio

tests due to the small sample size and were adjusted via Benjamini-Hochberg multiple testing corrections. Gene set enrichment analysis on sets of differentially abundant proteins were conducted via the R package *clusterProfiler*[60]. More specifically, enrichments were based on the gene ontology of proteins and focused on biological processes. Significance was based on the 'optimized FDR q-values' calculated within *clusterProfiler*. Protein lists were converted to Entrez-ids via org.Mm.eg.db for mice[61] and org.Hs.eg.db for humans[62] (version 3.18.0 for both). All other data were analyzed via Welch's t-test (2 groups) or F-test (3- or more groups) to account for any inequality of variances.

The differential abundance of metabolites was assessed using MetaboAnalyst 5.0 and 6.0. Datasets were manually filtered to remove redundancy, imported into MetaboAnalyst and Log10 transformed, analyzed via ANOVA (2+ groups) or t-test (2-groups, unequal variance), and exported for visualization in Graphpad Prism (v10) or Morpheus (https://software.broadinstitute.org/morpheus/).

Trends in compound hydrophobicity and recovery under modified forms of the AMW extraction procedure were assessed in the non-redundant data subset using cheminformatics. 242 of the 252 compounds detected by metabolomics were mapped to human metabolome database (HMDB) or PubChem IDs. Identifiers are unavailable for ten lipids in the dataset and these compounds were excluded. The set of 242 mapped compounds was subsequently filtered to remove redundancy, defined here as cases where the same compound was detected and quantified using two or more methods. This processing step avoids overrepresentation of a subset of chemicals (one-to-many relationships) that would otherwise bias the cheminformatics analysis. A total of 47 of 242 cases of redundancy were identified. These issues were resolved by including data from a single method that yielded the highest quality measurements for a given compound. Specifically, measurements were included for the method that minimized the coefficient of variation (CV) of pooled quality control (QC) samples. These processing steps resulted in a dataset of 195 compounds for understanding compound property-recovery relationships. The HMDB ID or PubChem ID was used to obtain the SMILES (Simplified Molecular Input Line Entry System) via a web-scraping routine implemented using the Python request library. The octanol-to-water coefficient (LogP) was predicted from SMILES descriptors using the python RDkit library. For each compound, the Log2 fold change (Log2 FC) of the mean pool size ($n = 3$) in the experimental condition (AMW25, AMW30, AMW35, AMW40, AMW45, AMW50, AMW55, AMW60) was calculated with respect to the control condition (AMW20). Trends in the Log2 FC and the LogP were analyzed by computing Spearman's rank correlation coefficient (non-parametric. Correlation coefficients were calculated using the python scipy.stats module. All cheminformatics scripts were developed using Python v3.9.7.

## Data availability
Metabolomics raw data generated in this study have been deposited in the MassIVE database under accession code MSV000095011 [https://massive.ucsd.edu/ProteoSAFe/dataset.jsp?task=b204220ef95c400393f834ff70471b5a]. Proteomics raw data have been deposited on ProteomeExchange under accession code PXD053052. Source data are provided with this paper. The source data underlying all figures included in the main text and supplementary materials are included in the Source Data File. Source data are provided with this paper.

## Code availability
Custom R scripts used for statistical analyses of the proteomics data[63] are available via Zenodo [https://doi.org/10.5281/zenodo.12191111].

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

## Acknowledgements

We would like to thank Drs. Russell Jones, Sara Nowinski, Evan Lien, and Nick Burton for their helpful suggestions and feedback, and Drs. Drew Jones and Mirela Berisa for their critical evaluation of this work. We also thank Megan Gendjar and Lisa DeCamp for their experimental assistance, and Drs. Matt Steensma and Carrie Graveel for providing HEK cells and tissue culture reagents. This work was supported by the VAI Metabolism & Nutrition (MeNu) Program, VAI Core Technologies and Services [Mass Spectrometry (RRID:SCR_024903) and Bioinformatics and Biostatistics (RRID:SCR_024762)], and grant 5T32CA251066-03 (RJH; PI: Peter A. Jones) from the National Cancer Institute (NCI). Figure 1a, Fig. 3a, and Supplementary Fig. 9a were created with BioRender.com and released under a Creative Commons Attribution-NonCommercial-NoDerivs 4.0 International license (agreement number ZI26XG37JO, FO26XG37L3, and TH26XG37MC, respectively).

## Author contributions

Conceptualization: R.J.H., M.T.S.H., R.D.S. Methodology: R.J.H., M.T.S.H., C.D. Capan, Z.B.M., A.E.E., C.N.I., C.D. Castello, A.B.J., M.P.V., H.L., R.D.S. Formal Analysis: R.J.H., M.T.S.H., C.D. Capan, Z.B.M., E.W., M.P.V., R.D.S. Investigation: R.J.H., M.T.S.H., C.D. Capan., A.E.E., C.N.I., A.B.J., M.P.V., R.D.S. Resources: H.L., R.D.S. Visualization: R.J.H., M.T.S.H., Z.B.M., E.W., M.P.V., K.W. Supervision: R.D.S. Project Administration: M.L.E.G., K.W., R.D.S. Funding acquisition: R.D.S. Writing—original draft: R.J.H., M.T.S.H., M.L.E.G., R.D.S. Writing—review & editing: R.J.H., M.T.S.H., C.D.Capan, Z.B.M., E.W., M.P.V., A.E.E., C.N.I., C.D. Castello, A.B.J., M.L.E.G., K.W., H.L., R.D.S.

## Competing interests

The authors declare no competing interests.
