## [Transparent Peer Review file · Nature Communications]

A diverse proteome is present and enzymatically active in metabolite extracts

Corresponding Author: Dr Ryan Sheldon

Figures originally included in the author's rebuttal have been redacted from this file.

Version 0:

Reviewer comments:

Reviewer #1

(Remarks to the Author)

This is an intriguing paper. It is clear that there are proteins in metabolomic extracts. I am less convinced that they are metabolically active, though there is at least some data suggesting it is feasible if the sample extracts are diluted to 50% water or more.

The authors present an approach whereby a sample is extracted in AMW20 (20% Water in acetonitrile/methanol). Upon dilution of this in additional water, metabolic transformations appear to begin, with increasing apparent activity with increasing water content. There is evidence for many metabolic enzymes present in the extract, as revealed by proteome analysis. There is also evidence of metabolic conversion of D5 to D4 glutamate, in a manner which is difficult to explain without invoking metabolic activity. Evidence is additionally presented that this may be mediated through Glutathione metabolic reactions.

I find all of this to be of interest, and plausible, but I do wonder whether the experiment is relevant - there is no apparent metabolic activity at AMW20 (fig 1F). How many labs would dilute their organic extracts with water prior to injection? I guess I could envision a lab doing so to improve reverse phase chromatography retention/binding, but can come up with no other obvious reason. That is, should the conclusions of this paper be rather 'don't dilute your 20% aqueous extracts in water?' and 'don't extract in less than 80% organic'?

There is certainly evidence of some enzymatic activity, but it also seems plausible that some of the reactions are non-enzymatic, and these alternative explanations are frequently not mentioned as potential mechanisms for the observed data.

Figure 2b - the impact of filtration seems quite small.

Figure 3a, AADAT. Why is the 20 so far from zero?

Fig 3 - why not show the MS/MS data to confirm the structures proposed in c and f?

Fig 5a - you see drift in AMW20+F, which should be the most non-enzymatic situation possible. You also see drift in AMW50+F, which should be mostly enzyme free, in PC2, away from the AMW20-F sample. It is difficult to interpret the relative import of this drift in PCA space as compared to the AMW50 samples, which you go on to further highlight. You show an example of metabolites that are changing, but how many change, how many are stable?

Fig 5d-g - does this represent AMW50?

"Protein carryover from sample types with metabolic enzymes"... I don't think 'protein carryover' is the correct term here - rather solubilized protein maybe? Carryover is generally used by analytical chemists to refer to sample to sample artifacts - i.e. carryover from one injection to the next. Carryover from the pipette has contaminated the next sample.

" α KG was not significantly different than control with AOA treatment. We interpret this as an indication of the presence of a non-AOA inhibitable deamination..." Why is this not evidence that the conversion is non-enzymatic? "Nitrogen labeling was

found only in aspartate (Figure 3h), providing clear evidence of glutamate aspartate transaminase activity. Collectively, these results provide clear evidence of enzymatic activity and metabolite interconversions in resuspended metabolomics extracts". I do not think that a lack of ^{15}N labelling in other amino acids supports the claim that Glu-Asp enzymatic claim.

"Consistent with elevated D4/D5 Glu, we also observed GOT1 and GOT2 protein in muscle" - was GOT1/2 also absent from the other sample types?

line 279: "with a methyl group (CH₂) neutral loss" - i think you mean methyl group addition.

line 306: "AMW metabolite extraction with 3 kDa filtration would simultaneously improve compound recovery" why would this improve recovery? This doesn't make much sense to me.

line 319: "suggesting that these compounds are lost through a protein-mediated mechanism in AMW50 extracts" - an alternate explanation is that you are seeing increased amino acids through proteolysis in the 3kDa filter retentate, releasing amino acids into solution, and being measured as increased concentrations. A similar mechanism could explain increased nucleotides (RNA/DNA degradation).

Reviewer #2

(Remarks to the Author)

In this manuscript, House et al used quantitative proteomics to systemically characterize the residual proteins in metabolite extracts. They showed that residual enzyme activity causes post-extraction metabolite interconversions, and it is prevalent in different metabolomics extraction approaches. Finally, the authors also demonstrated that a post-extraction filtration approach can help alleviate the problem. In short, this manuscript addressed a fundamental but neglectable issue of how residual proteins in metabolite extracts affect metabolite detection in a metabolomics study. Overall, I enjoyed reading this paper. The manuscript is well-organized and well-written. I also like the stable isotope labeling experiments to demonstrate the effects of enzymes on the conversion of glutamate and -KG. The major concern is that this manuscript is mainly limited to analytical chemistry, but it may not be of general interest to readers of Nature Communications. I would like to suggest some additional analyses/experiments to broaden the applicability of this work. I hope these can improve the quality and significance of the manuscript properly. If any comments are difficult to address, please feel free to discuss them.

1. The first thing I would suggest is to make these datasets easily reusable for common users. This will be a good way to increase the significance of the discovery and broaden the applicability of these results. Specifically, can the authors create a table (ideally on a web server, but an Excel table in the Supplementary Data is also good) to list which metabolic features/metabolites are altered in filtered or unfiltered extracts? What are the corresponding extraction solvents, protocols, and sample types? Based on this table, how can readers efficiently search for metabolites or metabolic features that may be affected by residual enzyme activity in their systems? Then they can address them in the subsequent analysis.

2. Fig 3a. It seems that only GOT1 was enriched in AMW50, while most of the enzymes were decreased in AMW50 compared to AMW20. I would like the authors to explain this a little bit.

3. Fig 3e. I noticed that the error bar of D4- -KG is very large. Can the authors repeat this conclusion? In addition, I would like to suggest that the authors use absolute quantification data instead of using relative peak area. The value of labeled D5-Glu should be close to the sum of D4-Glu and D4-alpha-KG. Similar results should also be obtained in the filtered and inhibitor groups.

4. P5 L205. I am glad to see that the authors tested several extraction procedures and different sample matrices in Figure 4. However, some common procedures, e.g. additional incubation at -20 °C after extraction (DOI: 10.1073/pnas.0812874106), freeze-thaw cycle with liquid nitrogen for cell disruption (DOI: 10.1007/s11306-015-0809-4), may also contribute to protein precipitation. These procedures are also common in the metabolomics field but were not tested here. Can the authors evaluate these extraction procedures whether affect the residual proteins and their activity?

5. Fig 5a-c. I was convinced that the residual protein activity affects the metabolites in extracts before LC-MS detection. The authors use PCA to demonstrate the global differences and show some interesting examples, but a quantitative characterization is welcome. For example, how many metabolites (the extent level) are affected by the residual protein activity? How much variation in their experiments can be explained by the residual proteomics data? One of the possible ways I came up with is that construct the regression curve between proteomics data and PC1 ($\text{PC1} \sim \text{proteomics data} + \text{time}$), where R² can be used to evaluate its explained variance.

6. P7 L278. The authors use the MS2 spectrum and the purified standard to validate the identification of S-methyl glutathione. That's great. The only concern is that I observed about 0.2 min RT error in Fig. S11. This error is a bit large. I think it would be much better if the authors could run the standard and sample in parallel.

7. P8 L322. The authors showed several examples to demonstrate the increased recovery. Like comment 3 above, these are qualitative conclusions. I would suggest that the authors add some quantitative results to answer how many recoveries can be increased after 3 kDa filtration.

Minors:

1. Fig 1c. I am not very clear about the meaning of the y-axis "the logP vs. log2FC (vs. AMW20)". Can the authors explain it in more detail?
2. Fig 1c. I guess "rs value" means the correlation coefficient. Please describe it in the legend.
3. P4 L177. I prefer to show the EIC of Figures 3d and 3g in the supplementary data as well. It would be clearer about the peak area differences.
4. Fig 3g-h. It would be great to show the result of the filtration data.
5. Fig 4a-b. Can you show the time-PC1 points for other groups (AMW20, AMW20+F, AMW50+F) in the Supplementary Data?
6. The resolution of Fig 4b is low in the manuscript I received. Please replace it with a high-resolution plot.

Reviewer #3

(Remarks to the Author)

The authors investigated the proteomic activity in metabolomics samples, and they provided many important results that metabolomics researchers need to know. I believe that this study is very important for our metabolomics community. The way to prevent the enzymatic activity in metabolite extractions is very simple, i.e., use of 3K-protein filter. The most significant concern I have regarding this paper is, where is the novelty? Many metabolomics researchers have considered the activity of proteins remaining in metabolite extracts and have conducted deproteinization processes. Some researchers will say that this paper merely objectively demonstrates the significance of deproteinization, and it can be argued that these results alone lack novelty in the realm of science. Furthermore, as the authors themselves mention, the solvent composition aside from protein filtering depends on the sample composition and should be optimized depending on the type of cells, blood, or tissue. Therefore, if the authors recognize the importance of their paper and aim to submit it to a high-impact venue like a Nature sister journal, they should discuss how such a protocol can be made known to metabolomics researchers and how a system can be formed as a consortium to achieve reproducibility across studies. Additionally, it is considered that a type of article such as correspondence would be preferable. Another important point is the complete lack of discussion on the impact on complex lipids. For instance, how much protein remains in the lower phase fraction of Bligh & Dyer, and to what extent is that protein activity preserved? If there were data on this, it could significantly increase the impact of this paper.

Author Rebuttal letter:

AUTHOR RESPONSE TO REVIEWER COMMENTS

Reviewer #1 (Remarks to the Author):

This is an intriguing paper. It is clear that there are proteins in metabolomic extracts. I am less convinced that they are metabolically active, though there is at least some data suggesting it is feasible if the sample extracts are diluted to 50% water or more.

The authors present an approach whereby a sample is extracted in AMW20 (20% Water in acetonitrile/methanol). Upon dilution of this in additional water, metabolic transformations appear to begin, with increasing apparent activity with increasing water content. There is evidence for many metabolic enzymes present in the extract, as revealed by proteome analysis. There is also evidence of metabolic conversion of D5 to D4 glutamate, in a manner which is difficult to explain without invoking metabolic activity. Evidence is additionally presented that this may be mediated through Glutathione metabolic reactions.

I find all of this to be of interest, and plausible, but I do wonder whether the experiment is relevant - there is no apparent metabolic activity at AMW20 (fig 1F). How many labs would dilute their organic extracts with water prior to injection? I guess I could envision a lab doing so to improve reverse phase chromatography retention/binding, but can come up with no other obvious reason. That is, should the conclusions of this paper be rather 'don't dilute your 20% aqueous extracts in water?' and 'don't extract in less than 80% organic'?

Author response. Thank you for this comment. It is very important that we adequately describe the premise of our approach to better frame the relevance and impact of our results. We thank you for giving us the opportunity to do so. There are a few points we would like to make to adequately address this concern.

First, based on the reviewer's question "How many labs would dilute their organic extracts with water prior to injection?", we fear that key rationale was effectively conveyed in the original draft. We did not dilute extracts in water and then inject for LCMS, but rather added water during extraction to demonstrate the effects of water addition on metabolomics readouts. These extracts were then dried and resuspended in water for LCMS. Note the same tissue equivalents were injected on-column in every condition. The goal, which is now better explained in the first paragraph of the results, was to improve polar metabolite recovery. In the revised version of the manuscript, we have moved data previously in supplemental Figure S1 to current Figure 1 to demonstrate the striking, metabolome-wide effects of extraction water content. It is in these dried and resuspended extracts where enzymatic activity was observed, as only at this point was the D5 glutamate added.

This is very relevant to the field at large. While AMW20 is certainly among the most common metabolite extraction approaches in the literature, it is by no means exclusive. In the discussion, we have highlighted numerous published protocols that use >20% water in organic solvent extracts. Some reach as high as 70% water. More concerning is the frequently sparse attention and under-reporting of metabolite extraction details in primary research papers. Of course, a systematic and comprehensive review of this issue cannot be completed here, but our paper raises serious concerns about research reproducibility imposed by inadequately controlled and detailed extractions. When one considers the contribution of sample water content to the total extract water content, this becomes more alarming still. Our results in Figure 1 and Figure S1 demonstrate a strong metabolite profile effect of water content, regardless of whether this is metabolic. For example, ATP detection doubles when water content increases from 20 to 30%. These data highlight the need to tightly control extraction water content. This topic is also discussed in the second paragraph of the discussion.

We appreciate the reviewers suggested alternative conclusion to not "dilute your 20% aqueous extracts in water" and to not "extract in less than 80% organic". However, our data demonstrate distinct advantages, namely improved recovery of many compounds like nucleotides, by increasing aqueous content in extracts that, when combined with 3 kDa filtration provide a superior method for polar metabolomics. Additionally, even in 20% aqueous solvents, we find abundant proteins in all sample types tested. This directly refutes the long-held but anecdotally supported dogma that organic solvent extraction removes proteins. This may pose several adverse analytical challenges, such as in-extract protein-metabolite interactions, that mask accurate phenotype detection even in the absence of bona fide catalytic activity.

There is certainly evidence of some enzymatic activity, but it also seems plausible that some of the reactions are non-enzymatic, and these alternative explanations are frequently not mentioned as potential mechanisms for the observed data.

Author response. This is an excellent point. We posit in the discussion (4th paragraph) that protein in metabolite extracts poses risks to accurate phenotype detection beyond catalytic activity. That some catalytic activity persists infers that proteins are not permanently denatured in metabolite extracts, as is commonly assumed. It stands to reason that other in-extract proteins retain or recover some degree of secondary and tertiary structure and "by extension" metabolite binding capacity. The scope of the biological protein-metabolite interactome is immense and not fully understood, and it likely exists in a different state in extraction solvents than in vivo. We have added the following statement to the discussion to address your point more specifically:

"It is likely that some of the time-dependent changes observed in protein replete AMW50 extracts (Figure 6) are due to protein-metabolite binding rather than catalytic activity. The extent of this in-extract proteome-metabolome interaction warrants further exploration."

Figure 2b - the impact of filtration seems quite small.

Author response. In the revised manuscript, these data are now shown in Figure 3b. Yes, 3 kDa filtration only removed roughly 30% of the total BCA signal. However, while the majority of BCA signal in extracts can be attributed to compounds less than 3 kDa in size, it indicates that ~3-4% of total liver protein is carried through to metabolite extracts. This is explained in the results as:

"BCA reagents also react with non-proteinaceous peptides, such as glutathione, which is abundant in the liver. To understand the fraction of BCA signal in AMW extracts arising from bona fide proteins, we filtered crude extracts through a 3 kDa filter. This resulted in a 20-30% decrease in calculated BCA protein content

compared to pre-filtration (Figure 3a-b), with the remaining signal assumed to arising from compounds less than 3 kDa. We subtracted the 3 kDa filter eluate BCA signal from total extract BCA signal and found that 2.7 and 4.1 μg per mg of liver tissue of proteins greater than 3 kDa are present in AMW20 and AMW50 extracts, respectively.

Figure 3a, AADAT. why is the 20 so far from zero?

Author response. We agree with the reviewer that this causes some confusion. The noted deviation of the mean from zero in the control (AMW20) was due to high variance leading to unequal weighting when plotted on a log₂ scale. Since our goal was to convey the presence of these proteins in extracts, rather than their relative abundances between groups, we have elected in the revised manuscript to change these to peak areas. These data are now in Figure S5.

Fig 3 - why not show the MS/MS data to confirm the structures proposed in c and f?

Author response. This is an excellent suggestion. We conducted this experiment using ddMS2 with narrow quadrupole isolation (0.4m/z) for each relevant glutamate isotopologue. This data is now included in Figure S6 and includes MS2 spectra corresponding fragment ion structures for each isotopologue. Specifically, note that these data in Figure S6b and S6c positively identify the deuterium on the α -amine carbon as the one exchanged by transamination as predicted by the model in new Figure 4a. We thank the reviewer for suggesting this approach.

Fig 5a - you see drift in AMW20+F, which should be the most non-enzymatic situation possible. You also see drift in AMW50+F, which should be mostly enzyme free, in PC2, away from the AMW20-F sample. It is difficult to interpret the relative import of this drift in PCA space as compared to the AMW50 samples, which you go on to further highlight. You show an example of metabolites that are changing, but how many change, how many are stable?

Author response. Thank you for this comment, and we agree it is imperative to establish the relative importance of time-dependent changes between groups. In the revised manuscript, we have done this in several ways.

1. We found all untargeted features with a significant non-zero slope (signal change over time) in any group and summarize these data as an UpSet plot in new Figure 6a and associated text in the Results. These data show that 89 features are significant between all groups, and likely represent compound-intrinsic drift. The next two most abundant groups are time-dependent changes unique to samples replete with protein, including 47 in AMW50, 24 in AMW50 and AMW20, and 20 unique to AMW20.

2. Using the PCA in Figure 6b, we calculated the accumulated Euclidean distance over time in each group for a semi-quantitative metric of time-dependent responsiveness. These data are in Figure S11b and described in text that AMW50 shows ~ 3 x greater magnitude of time-dependent shifts in AMW50 (13.1) versus other groups (AMW20: 5.0, AMW20F: 3.8, AMW50F: 4.8).

3. We added a heatmap (Figure S11a) of the top 100 time-dependent features (by p-value), which clearly demonstrate greater time-dependent effects in AMW50 versus other groups.

4. We plotted the top ten (by p-value) features with non-zero slope in at least one group in Figure S11c. These features depict the general increased magnitude and rate of change relative to the other groups.

5. The results section has been updated to reflect these additions and better establish the rationale for focusing on AMW50 changes in the rest of the figure.

Fig 5d-g - does this represent AMW50?

Author response. Yes, these are AMW50. In the former 5e-g (now figure 6f-h), AMW50F is also plotted as dashed lanes. However, in the case of β -glutamyl-glutamate (new Figure 6e-f), this compound was only detected in AMW50 samples. This is now specified in the figure legend.

"Protein carryover from sample types with metabolic enzymes"... i don't think 'protein carryover' is the correct term here - rather solubilized protein maybe? Carryover is generally used by analytical chemists to refer to sample to sample artifacts - i.e. carryover from one injection to the next. Carryover from the pipette has contaminated the next sample.

Author response. Great point. We have updated this sentence as: "If proteins are present in metabolite extracts from sample types with metabolic enzymes (e.g., cells or tissues) then it is plausible that these extracts are"

" ^{13}C was not significantly different than control with AOA treatment. We interpret this as an indication of the presence of a non-AOA inhibitable deamination..." why is this not evidence that the conversion is non-enzymatic?

Author response. This is a great question, and we have taken the following steps to address it:

1. We repeated the ^{13}C ^{15}N glutamate tracer experiment to include the filtered, protein removed condition (AMW50F). These data are now in Figure 4e. Protein removal completely prevents ^{13}C ^{15}N formation from ^{13}C ^{15}N glutamate, indicating it is a protein-mediated effect.

2. We used standard curves to get an absolute quantitation of Glu and ^{13}C in the sample (Figure S7 and 4e-f). The results demonstrate that the ^{13}C pool is only about 1% of the glutamate pool. Thus, it is possible that small amounts of catalytic activity on the large glutamate pool, possible occurring before AOA inhibition and/or incomplete AOA inhibition, could easily. The small amount of ^{13}C -glutamate and ^{15}N glutamate appearing in AOA samples, but being completely absent from AMW50F samples, supports this idea of low-level, minor catalytic activity in the AOA condition. We now explain this in the results as: "Interestingly, formation of [^{13}C] ^{15}N was prevented by 3 kDa filtration, but not AOA. The low ^{13}C to glutamate ratio (~1:100) likely explains this, as minor residual activity or kinetic mismatches (i.e., brief catalytic activity before AOA inhibition upon resuspension) would lead to rapid turnover of ^{13}C ."

"Nitrogen labeling was found only in aspartate (Figure 3h), providing clear evidence of glutamate aspartate transaminase activity. Collectively, these results provide clear evidence of enzymatic activity and metabolite interconversions in resuspended metabolomics extracts". I do not think that a lack of ^{15}N labelling in other amino acids supports the claim that Glu-Asp enzymatic claim.

Author response. This is a good point; we agree that these data alone are insufficient for this claim. In the revised manuscript we have now included absolute quantitation of Glu, Asp, and ^{13}C . This allows us to account for the total molar load of ^{15}N in the sample (Figure 6f). Indeed, the sum of ^{13}C ^{15}N Glu+ ^{15}N Glu+ ^{15}N Asp in AMW50 equals the amount of ^{13}C ^{15}N Glu that was unconverted in the AMW50F samples. So, the lack of labeling in other

amino acids plus this new data accounting for labeled nitrogen certainly support the glutamate-aspartate transamination. But, as the reviewer pointed out, we cannot ascribe true causation at this point, so we have toned-down the conclusion. It now states that this is "collectively suggesting prominent glutamate-aspartate transamination".

"Consistent with elevated D4/D5 Glu, we also observed GOT1 and GOT2 protein in muscle" - was GOT1/2 also absent from the other sample types?

Author response. We have added these data for each matrix to Figure S10.

line 279: "with a methyl group (CH_2) neutral loss" - i think you mean methyl group addition. Author response. Thank you for this comment. Here, we are intending to indicate that the neutral loss (i.e., an MS fragment that does not carry a charge and, thus, cannot be detected by MS) between the parent ion (unknown compound 320.0924) and the daughter ion with m/z consistent with GSH (306.0765) is 14.0156 Da. This neutral loss mass of 14.0156 is an accurate mass match to CH_2 , which would be expected from a methyl-group cleavage. We have modified the text in the results to better convey this information.

line 306: "AMW metabolite extraction with 3 kDa filtration would simultaneously improve compound recovery" why would this improve recovery? This doesn't make much sense to me.

Author response. We appreciate this comment, as it underscores that the original version of our manuscript did not adequately establish the premise of post-extraction water addition. The overhaul of Figure 1, described above, should alleviate this concern. In brief, in Figure 1 we demonstrate clear improvements in compound recovery, especially nucleotides, through extraction water addition. However, this condition also led to our

discovery of enzymatic activity in these extracts causes metabolic interconversions. So, here we sought to capitalize on the benefits of water addition while avoiding the protein-mediated consequences through filtration. We have modified this sentence to: "Given our observations that in-extract water addition improves polar metabolite detection (Figure 1), but also that it exposes extracts to post-extraction metabolic interconversions (Figures 2, 4, 6), we hypothesized that combining high-water AMW extraction with 3 kDa protein removal would improve polar metabolite coverage while mitigating the risks posed by post-extraction enzymatic activity"

line 319: "suggesting that these compounds are lost through a protein-mediated mechanism in AMW50 extracts" - an alternate explanation is that you are seeing increased amino acids through proteolysis in the 3kDa filter retentate, releasing amino acids into solution, and being measured as increased concentrations. A similar mechanism could explain increased nucleotides (RNA/DNA degradation).

Author response. Thank you for this comment. We agree that there are many possible consequences of protein retention in metabolite extracts, including protease and nuclease activity. This comment highlights the broad-reaching analytical challenges the field must now address given the protein presence in metabolite extracts we have demonstrated. Indeed, our proteomics data indicate the presence of several proteases/nucleases in extracts. We now specifically mention this possibility to the discussion: "Moreover, our proteomics data revealed the presence of several proteases and nucleases, which, if catalytically active, may lead to the liberation of amino acids and nucleosides from protein and RNA/DNA, respectively."

However, in this case, we do not believe it explains the metabolites that are decreased in AMW50 versus AMW50F for several reasons. First, if proteases/nucleases are in the retentate, then it stands to reason that they are also in the unfiltered samples, and any activity would be occurring in both. So, proteolytic activity would increase free amino acid content in both, so it seems unlikely to explain differences between these groups. If anything, one might expect the opposite effect (i.e., higher amino acids in AMW50 versus AMW50F), since proteins are not removed from AMW50, there would be longer incubation time of proteins and proteases in solution in AMW50 vs during filtration in AMW50F.
Reviewer #2 (Remarks to the Author):

In this manuscript, House et al used quantitative proteomics to systemically characterize the residual proteins in metabolite extracts. They showed that residual enzyme activity causes post-extraction metabolite interconversions, and it is prevalent in different metabolomics extraction approaches. Finally, the authors also demonstrated that a post-extraction filtration approach can help alleviate the problem. In short, this manuscript addressed a fundamental but neglectable issue of how residual proteins in metabolite extracts affect metabolite detection in a metabolomics study. Overall, I enjoyed reading this paper. The manuscript is well-organized and well-written. I also like the stable isotope labeling experiments to demonstrate the effects of enzymes on the conversion of glutamate and α -KG. The major concern is that this manuscript is mainly limited to analytical chemistry, but it may not be of general interest to readers of Nature Communications. I would like to suggest some additional analyses/experiments to broaden the applicability of this work. I hope these can improve the quality and significance of the manuscript properly. If any comments are difficult to address, please feel free to discuss them.

Author response. We thank the reviewer for their overall positive assessment of our work. We also greatly appreciate the helpful suggestions that were provided to enhance the applicability to a broader audience, which we have into the revised manuscript.

The first thing I would suggest is to make these datasets easily reusable for common users. This will be a good way to increase the significance of the discovery and broaden the applicability of these results. Specifically, can the authors create a table (ideally on a web server, but an Excel table in the Supplementary Data is also good) to list which metabolic features/metabolites are altered in filtered or unfiltered extracts? What are the corresponding extraction solvents, protocols, and sample types? Based on this table, how can readers efficiently search for metabolites or metabolic features that may be affected by residual enzyme activity in their systems? Then they can address them in the subsequent analysis.

Author response. Thank you for this suggestion, and we agree that our data will be a great resource for the community. To this end, in addition to the proteomics data that were provided in the original submission, we have now included supplementary files (File S1, containing proteomics data, and File S2, containing metabolomics data) of metabolomics

data related to Figures 1 (water titration metabolomics), 5 (multi-matrix metabolomics), and 7 (3 kDa filtration metabolomics). We have also will make accession numbers of raw LCMS files available via proteomeXchange prior to publication. Moreover, we have included our internally curated, standard-verified compound lists (including RT) to enable community analysis of our data. Finally, in the spirit of community usefulness, though beyond the scope of the present manuscript, we are in the process of preparing a protocol manuscript for the extraction modalities used in this paper, including the water addition and 3 kDa features novel to this manuscript. It is our intention to pursue peer-reviewed publication of the protocol once the present manuscript is published.

Fig 3a. It seems that only GOT1 was enriched in AMW50, while most of the enzymes were decreased in AMW50 compared to AMW20. I would like the authors to explain this a little bit.

Author response. This is an excellent point. We included these plots in the original version to emphasize those transaminases that were detected in metabolite extracts. However, as is apparent from our analysis in Figure 3, the abundance of individual proteins is highly dependent on extraction water content. We do not believe enzymatic activity is driven solely by protein abundance per se, but also by a host of factors including cofactor and substrate availability, ionic strength of the resuspended sample, protein folding, and likely other factors. To de-emphasize the importance of differential transaminase abundance, we have converted these plots to raw peak area (instead of \log_2FC), which demonstrates that these proteins are indeed present in each condition. We have also elected to move these plots to the supplementary files (Figure S5), as the connection of these to demonstrated enzymatic activity is only correlative. The protein must be present for activity to occur, but we cannot definitively assign the activity to a specific enzyme. This does not dilute the impact of Figure 4, as metabolic interconversions are ceased with protein removal, and mostly prevented with transaminase inhibition.

Fig 3e. I noticed that the error bar of D4- α -KG is very large. Can the authors repeat this conclusion? In addition, I would like to suggest that the authors use absolute quantification data instead of using relative peak area. The value of labeled D5-Glu should be close to the sum of D4-Glu and D4- α -KG. Similar results should also be obtained in the filtered and inhibitor groups.

Author response. This is an excellent observation and suggestion. The answer to the high α -KG variability is apparent with the newly added quantitation data that the reviewer suggested. Based on this comment, we used standard curves to assess absolute concentration of glutamate, α -KG, aspartate. From this, we can conclude that the α -KG pool is roughly 1% of the glutamate pool. Thus, it is possible that small amounts of catalytic activity on the large glutamate pool, possibly occurring before AOA inhibition and/or incomplete AOA inhibition, could easily. The small amount of ^{13}C -glutamate and ^{15}N glutamate appearing in AOA samples, but being completely absent from AMW50F samples, supports this idea of low-level, minor catalytic activity in the AOA condition. We now explain this in the results as: Interestingly, formation of $[^{13}C_5]\alpha$ -KG was prevented by 3 kDa filtration, but not AOA. The low α -KG to glutamate ratio (~1:100) likely explains this, as minor residual activity or kinetic mismatches (i.e., brief catalytic activity before AOA inhibition upon resuspension) would lead to rapid turnover of α -KG. Further, we repeated the $^{13}C^{15}N$ glutamate tracer experiment to include the filtered, protein removed condition (AMW50F). These data are now in Figure 4e. Protein removal completely prevents ^{13}C α -KG formation from $^{13}C^{15}N$ glutamate, indicating it is a protein-mediated effect and supporting an incomplete inhibition by AOA.

4. P5 L205. I am glad to see that the authors tested several extraction procedures and different sample matrices in Figure 4. However, some common procedures, e.g. additional incubation at $-20^\circ C$ after extraction (DOI: 10.1073/pnas.0812874106), freeze-thaw cycle with liquid nitrogen for cell disruption (DOI: 10.1007/s11306-015-0809-4), may also contribute to protein precipitation. These procedures are also common in the metabolomics field but were not tested here. Can the authors evaluate these extraction procedures whether affect the residual proteins and their activity?

Author response. Thank you for this suggestion. We have conducted this experiment as you suggested, and further included overnight incubation at $80^\circ C$ before or after water addition (inducing a freeze-thaw cycle) as a means to remove proteins or otherwise prevent enzymatic activity. Remarkably, prolonged extract incubation at $80^\circ C$ does not affect the proteomic composition of the extracts nor does it prevent enzymatic activity in AMW50 extracts. These data are now a new supplemental figure (Figure S9), which, when combined with the multi-extraction modalities and sample types (Figure 4), help convey the ubiquitousness of protein retention and the need for their removal through other means

(e.g., 3 kDa filtration).

5. Fig 5a-c. I was convinced that the residual protein activity affects the metabolites in extracts before LC-MS detection. The authors use PCA to demonstrate the global differences and show some interesting examples, but a quantitative characterization is welcome. For example, how many metabolites (the extent level) are affected by the residual protein activity? How much variation in their experiments can be explained by the residual proteomics data? One of the possible ways I came up with is that construct the regression curve between proteomics data and PC1 ($PC1 \sim \text{proteomics data} + \text{time}$), where R^2 can be used to evaluate its explained variance.

Author response. This is a very useful comment that has led us to expand our description of data this experiment in the results. Reviewer #1 had a similar comment, and some of our response to that comment is duplicated here:

1. We found all untargeted features with a significant non-zero slope (signal change over time) in any group and summarize these data as an UpSet plot in new Figure 6a and associated text in the Results. These data show that 89 features are significant between all groups, and likely represent compound-intrinsic drift. The next two most abundant groups are time-dependent changes unique to samples replete with protein, including 47 in AMW50, 24 in AMW50 and AMW20, and 20 unique to AMW20. When considering that filtration removes protein from the extract, these data address the reviewer's question of "How much variation in their experiments can be explained by the residual proteomics data?"

2. Using the PCA in Figure 6b, we calculated the accumulated Euclidean distance over time in each group for a semi-quantitative metric of time-dependent responsiveness. These data are in Figure S11b and described in text that AMW50 shows ~ 3 x greater magnitude of time-dependent shifts in AMW50 (13.1) versus other groups (AMW20: 5.0, AMW20F: 3.8, AMW50F: 4.8).

3. We added a heatmap (Figure S11a) of the top 100 time-dependent features (by p-value), which clearly demonstrate greater time-dependent effects in AMW50 versus other groups.

4. We plotted the top ten (by p-value) features with non-zero slope in at least one group in Figure S11c. These features depict the general increased magnitude and rate of change relative to the other groups.

5. The results section has been updated to reflect these additions and better establish the rationale for focusing on AMW50 changes in the rest of the figure.

6. P7 L278. The authors use the MS2 spectrum and the purified standard to validate the identification of S-methyl glutathione. That's great. The only concern is that I observed about 0.2 min RT error in Fig. S11. This error is a bit large. I think it would be much better if the authors could run the standard and sample in parallel.

Author response. As requested, we have repeated validation experiments of Glu-Glu an MeGSH using standard spiked into experimental replicates. These data are now in supplemental Figure S12c and Figure S13d, respectively. The 0.2 min RT differences between experimental samples and standards noted by the reviewer in the original data was due to a combination of differences in sample matrix (i.e., liver extract versus neat standard) and a new LC column between runs.

7. P8 L322. The authors showed several examples to demonstrate the increased recovery. Like comment 3 above, these are qualitative conclusions. I would suggest that the authors add some quantitative results to answer how many recoveries can be increased after 3 kDa filtration.

Author response. Thank you for this comment, and in response we have added quantitative results to these data, which are in Figure 7 and S15 in the revised manuscript. Additionally, our overhaul of Figure 1 better demonstrate the quantitative impacts of water addition on polar metabolite recovery, which better sets the premise for this figure where we seek to evaluate 3kDa filtration to preventing unwanted metabolic interconversions from protein presence. A summary of the changes are:

1. Inclusion of volcano plots of relevant pairwise comparisons (AMW50F/AMW50 and AMW50F/AMW20, Figure 7; AMW50/AMW20 and AMW20F/AMW20, Figure S15) to highlight the magnitude of fold-changes. Points on these plots are colored based on cluster designations in Figure 7a to facilitate interpretation.

2. Inclusion of nucleotide bar plots in Figure S15b to demonstrate that the

increased nucleotide recovery with water addition from figure 1 was repeated in this experiment and not negatively affected by filtration.

3. Inclusion of select polar lipids to demonstrate that 3 kDa filtration specifically in AMW50 leads to depletion of these compounds. This could be perceived as a negative data because it shows a limitation of the new method we are proposing but we hope the reviewer will appreciate our effort to maintain full transparency. The loss of these lipids is also mentioned in the discussion as a limitation. These data further demonstrate the strong effect that extraction conditions can have on the detected metabolome, underscoring the need for more complete, detailed method reporting.

4. Peak area tables from this experiment are now included as Supplementary File 1, and raw MS files will be deposited on proteomeXchange and accession numbers for the raw data provided with the published version of this manuscript to encourage community use of our data.

Minors:

1. Fig 1c. I am not very clear about the meaning of the y-axis (the logP vs. log2FC (vs. AMW20)). Can the authors explain it in more detail?

Author response. The intention with these plots was to demonstrate that as extraction water content increases, there is not a concomitant correlation with analyte hydrophobicity (logP). If analyte hydrophobicity was solely responsible for analyte response to extraction water content, then we would have expected this to become increasingly negative (i.e., high water content improves recovery of more hydrophilic, negative logP, compounds). Instead, we saw this correlation weaken and become less-positive as water content increased. This indicated that non-compound intrinsic factors were driving the metabolite response to water content. We have rearranged the presentation of Figure 1 to help with this flow, and have expanded text in the results to better explain this data:

Using these predicted values and the fold-change with water addition relative to AMW20, we hypothesized an inverse correlation between LogP and the Log2FC of increasing water content relative to AMW20. Consistent with this, a significant inverse correlation was observed beginning at AMW30 (Figure 1h). However, instead of becoming more negative with increasing water content, this correlation weakened and became less negative (Figure 1j; Figure S2). These data, summarized as a function of water content in Figure 1j, collectively indicate that hydrophobicity alone does not fully predict compound recovery from increased extraction water content and further suggest the influence of compound extrinsic factors in this response.

2. Fig 1c. I guess the r value means the correlation coefficient. Please describe it in the legend.

Author response. Thank you, this is now described in the figure legend.

3. P4 L177. I prefer to show the EIC of Figures 3d and 3g in the supplementary data as well. It would be clearer about the peak area differences.

Author response. Representative extracted ion chromatograms for each relevant glutamate isotopologue are now included in Supplemental Figure S6.

4. Fig 3g-h. It would be great to show the result of the filtration data.

Author response. Excellent suggestion. We repeated this experiment with the AMW50F condition included and the data are presented in the new Figure 4. This turned out to be critical, because it revealed that AOA inhibition is not absolute. Whether residual activity in AOA is due to incomplete inhibition or some non-AOA inhibitable mechanism isn't clear, but the addition of the filtration condition to this experiment confirms that it is a protein-mediated mechanism nonetheless.

5. Fig 4a-b. Can you show the time-PC1 points for other groups (AMW20, AMW20+F, AMW50+F) in the Supplementary Data?

Author response. We have included these as a reviewer only figure below, however we elected not to add these to the manuscript for the following reasons. As can be seen from the all-group PCA in Figure 6b, time (i.e., injection order) is the primary source of variance within each group, and individual group PCA scores are agnostic to the global variance in the combined dataset (i.e. X and Y scales are relative and arbitrary within each plot), so

plots show similar time-dependent stratification in each group. This leads one to the erroneous conclusion that within group, time-dependent drift is equivocal. However, as described in our response to the Reviewer's Comment #5 in the previous section, time-dependent changes are much greater in magnitude in AMW50 than the other three groups. For these reasons, we believe that the additions of UpSet plots (Figure 6a), heatmap of time-dependent changes (Figure S11a), accumulated PCA Euclidean distance (Figure S11b), and line plots of the top 10 (by p-value) features with non-zero slope (Figure S11c) more accurately convey to the reader that there are time-dependent changes in all groups, but the scope and magnitude is markedly enhanced in AMW50. This, then, leads us to examine AMW50 more fully in the rest of Figure 6.

[Redacted]

6. The resolution of Fig 4b is low in the manuscript I received. Please replace it with a high-resolution plot.

Author response: The plot has been replaced by a higher resolution one.

Reviewer #3 (Remarks to the Author):

Reviewer 3 Comment 1: The authors investigated the proteomic activity in metabolomics samples, and they provided many important results that metabolomics researchers need to know. I believe that this study is very important for our metabolomics community.

Author response. Thank you for your overall positive assessment of our work. We fully agree that this is critical information for metabolomics community, and more broadly for any biomedical science or related discipline that utilizes metabolomics.

The way to prevent the enzymatic activity in metabolite extractions is very simple, i.e., use of 3K-protein filter. The most significant concern I have regarding this paper is, where is the novelty? Many metabolomics researchers have considered the activity of proteins remaining in metabolite extracts and have conducted deproteinization processes. Some researchers will say that this paper merely objectively demonstrates the significance of deproteinization, and it can be argued that these results alone lack novelty in the realm of science.

Author response. Thank you for giving us the opportunity to respond to this concern, as it is imperative that the novelty of this work is fully conveyed. Novelty is summarized here:

1. Metabolite extracts contain not just a few, but over a thousand proteins, that span sample and extraction types. Currently, widely accepted dogma is that organic solvent metabolite extraction rapidly quenches enzymatic activity, permanently denatures proteins, and completely remove them by precipitation, in turn leaving purified small molecules of interest in the soluble fraction. While proteins are certainly present in the insoluble fraction, this does not necessitate their absence from the metabolite fraction. This notion that metabolite extraction "crashes out" protein is heralded in introductory lectures, courses, and review articles on metabolomics. The data we present in the current manuscript directly refute this dogma and will need to be addressed by the community.

2. We agree with the reviewer that "Many metabolomics researchers have considered the activity of proteins remaining in metabolite extracts and have conducted deproteinization processes". However, we contend that:

a. To our knowledge any previous consideration of post-extraction enzymatic activity is centered on briefly maintained catalytic activity persisting immediately after initial quenching. Namely, formic acid has been evaluated as an extraction additive to more rapidly inactivate enzymes on extraction and prevent ATP hydrolysis. We have expanded text on this matter in the second paragraph of the discussion. Further, we note that post-extraction ATP hydrolysis has never been positively ascribed to enzymatic activity, rather such an effect is implied from readouts of ATP and ATP/ADP ratio in various conditions. ATP → ADP hydrolysis can also be spontaneous, as is evident by the presence of some ADP even in neat analytical standards of ATP. The concept that enzymatic activity can persist throughout extraction, drying, and resuspension is wholly new.

b. The "deproteinization process" used throughout the field is homogenization in organic solvent, which is the very same demonstrated unequivocally in the current paper to contain >1,000 proteins. To-date, the deproteinization process has been assumed to be complete, inferring this from the presence of proteins in the insoluble pellet, but ours is the

first attempt to characterize this systematically. In fact, what the field has thought to be a deproteinization process is in fact only a protein-reduction approach. We find that 3-4% of sample protein persists in the soluble fraction. As stated above, the current work represents a dogmatic shift in how protein behavior in organic solvents is viewed.

3. Moreover, now that we know that extracts are neither protein free nor are proteins fully denatured, we as a community must know contend with the broad reaching consequences this poses to analytical fidelity. The implications are vast and will take time to unveil. For example, in addition to catalytic activity, protein-metabolite binding in extracts may conceal metabolites from analysis (1) by retaining a fraction of the metabolome in the insoluble fraction (which is usually discarded) of the crude extract and/or (2) by allowing de novo protein-metabolite binding events in the extracted sample. When one considers that proteins are also components of biological phenotype expression, this raises the possibility that a true protein phenotype may present itself as a false-positive metabolomic phenotype.

4. In response to this paper merely objectively demonstrates the significance of deproteinization, we agree that deproteinization is critical, and our data provide novel insights into specific consequences of failing to do so. However, we also demonstrate that current universally accepted methods to accomplish deproteinization are, in fact, inadequate to this end. And, we provide a novel yet simple solution to this problem: 3kDa filtration. Our paper will force the community to reevaluate many long-held assumptions about the effectiveness of organic solvent deproteinization, and further consider how such assumptions may obscure true phenotype detection in metabolomics.

5. Additionally, we provide a metabolome-scale evaluation of extraction water content. The extreme sensitivity (many changes occurring with a mere 5% change in water content) complexity of the metabolomic responses to extraction water content (Figure 1B) highlight the need to both tightly control extraction variables and improve reporting of such details in manuscripts.

Furthermore, as the authors themselves mention, the solvent composition aside from protein filtering depends on the sample composition and should be optimized depending on the type of cells, blood, or tissue. Therefore, if the authors recognize the importance of their paper and aim to submit it to a high-impact venue like a Nature sister journal, they should discuss how such a protocol can be made known to metabolomics researchers and how a system can be formed as a consortium to achieve reproducibility across studies. Additionally, it is considered that a type of article such as correspondence would be preferable.

Author response. We appreciate this comment, and we fully agree that this needs to be disseminated to the broader community. This paper represents a paradigm shift, as described above, and the implications will take time to unpack. The first step, but not the only step, is the publication of a rigorous and thorough investigation, such as what we present here, published in a highly visible journal like Nature Communications. But completing the paradigm shift will require more than is possible with any single paper. To this end, we are taking steps to enhance the immediate impact and visibility of this manuscript. First, we have prepared a Comment piece at the invitation of editors of Nature Metabolism to discuss the implications of this work (we will share a preprint draft of this at the request of the reviewer) that will be submitted to that journal concomitantly with the resubmission of the present revision. Second, we are preparing a protocol manuscript to submit after publication of the present manuscript to further enable the community use of our approach. Further, we fully agree that a consortium to achieve reproducibility across studies is necessary to affect change. We believe that the current paper will provide the necessary rallying point for such a consortium to form.

Another important point is the complete lack of discussion on the impact on complex lipids. For instance, how much protein remains in the lower phase fraction of Bligh & Dyer, and to what extent is that protein activity preserved? If there were data on this, it could significantly increase the impact of this paper.

Author response. Thank you for this great suggestion. We have included DIA proteomics data for the Bligh-Dyer organic phase in Figure 5. To our surprise, this fraction contains over 1,800 proteins, 1,050 of which were unique to this extract type. GSEA of this fraction to the whole liver proteome is in Figure S8, and, like the other extraction modalities, reveals a similar enrichment of metabolic pathways. Purine/pyrimidine metabolic proteins appear to be particularly enriched in this fraction. This is now specified in the results. As these extracts when used for lipidomics are often dried and resuspended in solvent, like isopropanol, it seems unlikely that these proteins would be catalytically active. But

protein-lipid interactions may obscure phenotype detection by LCMS. Fully characterizing such effects is beyond the scope of this paper, which should be the focus of future efforts. To this end, we have added discussion to the third paragraph of the discussion:

Moreover, these proteins are present in common metabolite extraction modalities including AMW20, 80% methanol, and Bligh-Dyer, though specific protein composition varies in each. Remarkably, this was also true for the organic, lipid containing layer of Bligh-Dyer extracts. This fraction is commonly used for lipidomics, and the impacts of post-extraction lipid-protein interactions should be the focus of future studies.

Version 1:

Reviewer comments:

Reviewer #1

(Remarks to the Author)

The authors have responded satisfactorily to the comments. It is certainly an interesting set of results and should prompt good discussion amongst metabolomics practitioners.

Reviewer #2

(Remarks to the Author)

The authors did excellent work to address my concerns. I don't have any more questions.

Reviewer #3

(Remarks to the Author)

The authors have addressed the revision for my previous comments seriously and have made significant improvements to the manuscript. I believe this revised version is much improved. Therefore, I recommend this paper for publication in Nature Communications.
